# *Pseudomonas aeruginosa* SutA wedges RNAP lobe domain open to facilitate promoter DNA unwinding

Dingwei He[1,2,6], Linlin You[1,2,6], Xiaoxian Wu[1], Jing Shi[3], Aijia Wen[4], Zhi Yan[1], Wenhui Mu[1,5], Chengli Fang [1,2], Yu Feng [4,7✉] & Yu Zhang [1,7✉]

*Pseudomonas aeruginosa* (*Pae*) SutA adapts bacteria to hypoxia and nutrition-limited environment during chronic infection by increasing transcription activity of an RNA polymerase (RNAP) holoenzyme comprising the stress-responsive σ factor $\sigma^S$ (RNAP-$\sigma^S$). SutA shows no homology to previously characterized RNAP-binding proteins. The structure and mode of action of SutA remain unclear. Here we determined cryo-EM structures of *Pae* RNAP-$\sigma^S$ holoenzyme, *Pae* RNAP-$\sigma^S$ holoenzyme complexed with SutA, and *Pae* RNAP-$\sigma^S$ transcription initiation complex comprising SutA. The structures show SutA pinches RNAP-β protrusion and facilitates promoter unwinding by wedging RNAP-β lobe open. Our results demonstrate that SutA clears an energetic barrier to facilitate promoter unwinding of RNAP-$\sigma^S$ holoenzyme.

[1] Key Laboratory of Synthetic Biology, CAS Center for Excellence in Molecular Plant Sciences, Shanghai Institute of Plant Physiology and Ecology, Chinese Academy of Sciences, Shanghai, China. [2] University of Chinese Academy of Sciences, Beijing, China. [3] Department of Pathogen Biology, School of Medicine & Holistic Integrative Medicine, Nanjing University of Chinese Medicine, Nanjing, China. [4] Department of Biophysics, and Department of Pathology of Sir Run Run Shaw Hospital, Zhejiang University School of Medicine, Hangzhou, China. [5] Key Laboratory of Plant Stress Biology, State Key Laboratory of Cotton Biology, School of Life Sciences, Henan University, Kaifeng, China. [6]These authors contributed equally: Dingwei He, Linlin You. [7]These authors jointly supervised this work: Yu Feng, Yu Zhang. ✉email: yufengjay@zju.e.du.cn; yzhang@sippe.ac.cn

Recent studies discovered SutA as a new binding partner of *Pseudomonas aeruginosa* (*Pae*) RNAP[1,2]. The expression of SutA is induced under anaerobic survival conditions to help maintaining minimal transcription activity of *Pae* RNAP. Deletion of SutA causes global gene transcription changes, including substantially decreased transcription of genes encoding ribosomal protein and ribosomal RNA, and severe phenotypic defects, including reduced biofilm formation, virulence, and fitness[1]. Bioinformatic analysis suggests that SutA is present in most species in Alteromonadales and Pseudomonadales orders of Gamma-proteobacteria. SutA has no sequence homology to any other characterized proteins. Previous NMR study and secondary structural prediction suggested that SutA comprises a middle α-helix, a N-terminal D/E-rich region, and a C-terminal disordered tail[2]. Both the middle and N-terminal regions are required for its transcription activation activity on the promoter of ribosomal RNA *rrn*[2].

RNAP-β protrusion is likely the primary anchor site of SutA[2]. SutA interacts with both RNAP-σ$^A$ and RNAP-σ$^S$ holoenzymes and has greater effect on increasing transcription activity of the RNAP-σ$^S$ holoenzyme compared with RNAP-σ$^A$ holoenzyme[2]. Previous results of cross-linking and FeBABE cleavage assays suggested that SutA first associates with RNAP-σ$^S$, subsequently undergoes conformational change upon promoter dsDNA loading and unwinding, and eventually dissociates upon formation of the RNAP-promoter DNA open complex (RPo)[2]. However, due to the lack of structural information of SutA and SutA-RNAP complexes, the detailed interaction between SutA and RNAP is unknown and the molecular mechanism underlying SutA transcription activation remains elusive.

Promoter unwinding by bacterial RNAP is a multiple-step process[2–6]. First, RNAP recognizes the double-stranded promoter DNA (dsDNA) by making sequence-specific interactions with upstream promoter elements (the UP, −35, and/or extended −10 elements) and presents the downstream dsDNA on top of the main cleft; second, RNAP bends and melts the promoter dsDNA at the −10 element and accommodates the nucleotide at the −11 position of non-template strand DNA (NT-11A) in its cognate pocket; third, promoter unwinding propagates to the downstream of the −11 position and the nucleotides at other key positions (NT-7T, NT-6G, and NT + 2 G) are subsequently recognized and secured in respective protein pockets; lastly, the template ssDNA is loaded into and restrained in the active-center cleft ready for initiation of RNA synthesis[7].

The process of promoter dsDNA unwinding has to overcome several obstacles inside of RNAP, including: 1) the gate loop of RNAP-β lobe domain and the σ$_{1.2}$ domain that interact with each other to seal the upper main cleft[8], 2) the σ$_{1.1}$ domain located at the downstream dsDNA channel[9–11], and 3) the fork loop 2 and switch region 2 motifs that seal the lower main cleft[12]. Previous studies have reported cryo-EM structures of RNAP-promoter DNA intermediate complexes that were trapped by these obstacles during RPo formation, suggesting that they are the rate-limiting factors of promoter unwinding[7,12,13]. A collection of biochemical, structural, and computational data show that widening the main cleft by opening either clamp domain or lobe domain is the primary means to clear the obstacles for protomer loading and unwinding[7,14–17]. Not surprisingly, these obstacles are manipulated by macromolecular regulatory proteins, such as bacterial proteins DksA[13,18], TraR[7], and phage protein gp2[19], and small-molecular compounds, such as myxopyronin[12,16,20], lipiarmycin[21,22], and ppGpp[11], to accelerate or hinder the process of RPo formation through lowering or elevating energetic barriers imposed by these obstacles.

Compared with the above-mentioned transcription regulatory factors, SutA is unique in its large portion of disordered regions,

highly structural flexibility, and RNAP-σ$^S$ holoenzyme preference. To understand the molecular mechanism of SutA activation, here we determined cryo-EM structures of RNAP-σ$^S$ holoenzyme at 4.1 Å, SutA-bound RNAP-σ$^S$ holoenzyme at 3.1 Å (SutA-RNAP-σ$^S$, open lobe) and 3.9 Å (SutA-RNAP-σ$^S$, closed lobe), and SutA-bound RNAP-σ$^S$ promoter open complex (SutA-σ$^S$-RPo) at 5.8 Å. The structures reveal that SutA pinches RNAP-β protrusion with its RNAP-binding domain and wedges RNAP-β lobe open to facilitate promoter unwinding with its wedge loop and N-terminal D/E-rich region.

## Results

**SutA-RBD interacts with RNAP-β protrusion.** Previous study has indicated that RNAP-β protrusion is likely the anchor site of SutA[2]. We first used a yeast two-hybrid (YTH) assay to map SutA domains responsible for interaction with RNAP-β protrusion. The results confirmed the interaction between the full-length SutA and *Pae* RNAP-β protrusion previously identified by cross-linking experiments[2], and further showed the middle region (residues 57–90) of SutA comprising one precited α-helix and one β-strand is mainly responsible for the interaction (Fig. 1a, b). Therefore, we designated the region as the RNAP-binding domain of SutA (SutA-RBD). Although SutA-NTD does not interact with RNAP-β protrusion, it is essential for the transcription activation activity of SutA on RNAP-σ$^S$ holoenzyme (Fig. 1c, left panel and Supplementary Fig. 1e)[2]. In contrast, the short C-terminal tail of SutA contributes less to the transcription activation activity (Fig. 1c, left panel).

**The cryo-EM structure of RNAP-σ$^S$ holoenzyme.** As no structural information is available for bacterial RNAP- σ$^S$ holoenzyme, we reconstituted *Pae* RNAP-σ$^S$ holoenzyme and determined its structure through the single-particle cryo-EM approach (Supplementary Fig. 1a, b). Focused 3D classification resulted in a single group of RNAP-σ$^S$ holoenzyme particles, from which a cryo-EM map of the RNAP-σ$^S$ holoenzyme was reconstituted at resolution 4.1 Å (Supplementary Table 2 and Supplementary Fig. 2). The cryo-EM map shows strong continuous signals for all domains of σ$^S$ except for σ$^S_{1.1}$ (Fig. 2a and Supplementary Fig. 3a). The weak fractioned map signal of σ$^S_{1.1}$ domain in the main cleft of RNAP suggests its conformational heterogeneity and low occupancy. We compared the binding affinity of *Pae* RNAP core enzyme towards σ$^A_{1.1}$ or σ$^S_{1.1}$ by using a fluorescence polarization assay. The results showed that the σ$^A_{1.1}$ domain binds RNAP core enzyme with a Kd value of 220 nM, but the interaction between σ$^S_{1.1}$ domain and RNAP core enzyme is barely detectable under the same experimental condition (Fig. 2c), indicating that σ$^S_{1.1}$ binds to RNAP core enzyme much weaker compared with σ$^A_{1.1}$.

**The cryo-EM structure of SutA-RNAP-σ$^S$.** To explore how SutA interacts with RNAP, we reconstituted *Pae* SutA-bound RNAP-σ$^S$ complex and determined its cryo-EM structure (Supplementary Fig. 1c, d). Focused 3D classification reveals one major population of SutA-bound RNAP-σ$^S$ single particles, from which a cryo-EM map was reconstructed at a resolution of 3.1 Å, and one minor population of SutA-bound RNAP-σ$^S$ single particles, from which a cryo-EM map was reconstructed at a resolution of 3.9 Å (Supplementary Table 2 and Supplementary Fig. 4). Both maps show unambiguous signals for SutA near the RNAP-β protrusion and for all domains of σ$^S$ except for σ$^S_{1.1}$. The two maps exhibit different binding modes of SutA and distinct conformations of RNAP-β lobe, a structure module comprising RNAP-β2 and -βSI1 (also named as βi4), and accordingly the two structures were named SutA-RNAP-σ$^S$ (open lobe) and SutA-RNAP-σ$^S$ (closed lobe) (Fig. 3a–d and Supplementary Fig. 7a, b).

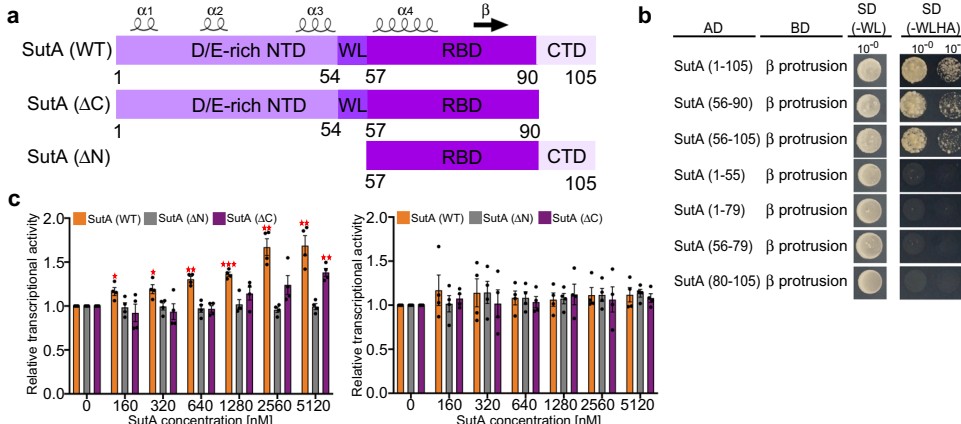

**Fig. 1 The requirement of SutA domains in RNAP interaction and transcription activation. a** SutA domain scheme annotated with predicted secondary structure features. NTD, N terminal domain; WL, wedge loop; RBD, RNAP-binding domain; CTD, C-terminal domain. **b** Interaction between the SutA domains and RNAP-β protrusion analyzed by the YTH assay. AD, activation domain of GAL4; BD, DNA-binding domain of GAL4. **c** SutA activates transcription of the RNAP-$\sigma^S$ holoenzyme (left) but has much less effect on RNAP-$\sigma^A$ holoenzyme (right) in a fluorescence-based in vitro transcription assay. Truncation of SutA-NTD (SutA (ΔN)) abolishes the activation activity of SutA. The data were presented as mean±S.E.M., n = 4. The individual data points were shown in black dots. Two-sided Student's t test was performed. *$p < 0.05$, **$p < 0.01$, ***$p < 0.001$, n = 4. Source data are provided as a Source Data file.

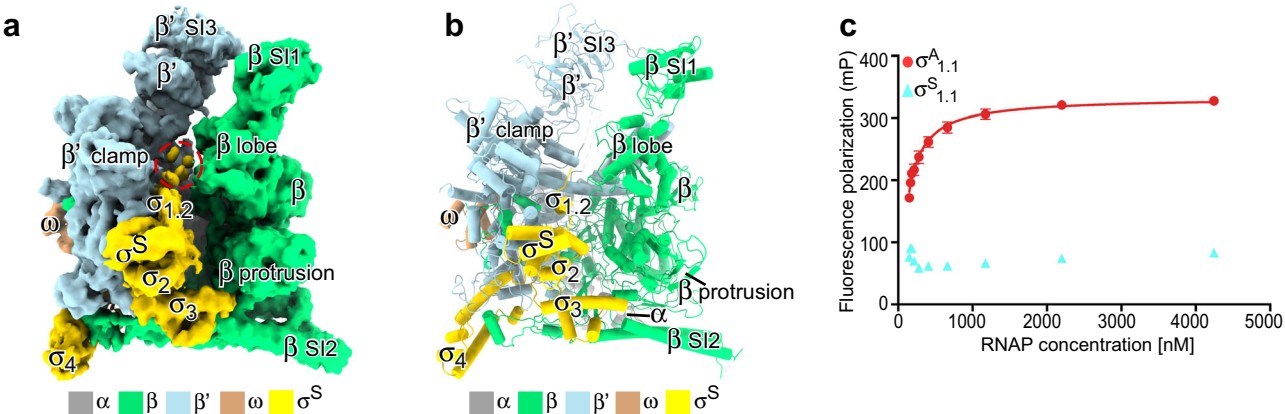

**Fig. 2 The cryo-EM structure of _Pae_ RNAP-$\sigma^S$ holoenzyme. a** The cryo-EM map for _Pae_ RNAP-$\sigma^S$ holoenzyme. RNAP subunits are colored as the color scheme. The $\sigma^S_{1.1}$ map is circled. **b** The structural model for _Pae_ RNAP-$\sigma^S$ holoenzyme ($\sigma^S_{1.1}$ is not modeled due to poor cryo-EM map signal around the region). **c** The results of fluorescence anisotropy assay showed the binding of $\sigma^A_{1.1}$ or $\sigma^S_{1.1}$ to RNAP. The data were presented as mean±S.E.M., n = 4. Source data are provided as a Source Data file.

In the structure of SutA-RNAP-$\sigma^S$ (open lobe), the RNAP-β protrusion is embraced by a patch of crescent-shaped map that could be assigned to SutA-RBD (Fig. 3a–b and Supplementary Fig. 3b), consistent with our YTH results and cross-linking data in a previous study[2]. The 3.1 Å cryo-EM map exhibits clear feature of an α-helix in the middle of the crescent-shaped map that fits well to the 20-residue predicted helical region (residues 57–76; Fig. 3a, f)[2]. The flanking residues of the middle helix were modeled into both ends of the crescent-shaped map (Supplementary Fig. 3b). One end of the crescent (residues 77–90) travels across the surface of the RNAP-β protrusion and approaches $\sigma_{3.1}$, four residues (81–84) of which likely form a 4-strand β sheet with RNAP-β protrusion (Fig. 3a, e). Intriguingly. The other end of the crescent (residues 54–57; wedge loop) fits into the narrow gap between RNAP-β protrusion and β lobe and leads the rest part of SutA-NTD to the main cleft (Fig. 3a and Supplementary Fig. 3b), where it likely adopts an assemble of multiple conformations and thereby does not show distinct map signals (Fig. 3a, g).

SutA-RBD occupies a hydrophobic groove on RNAP-β protrusion. Several hydrophobic residues of SutA (L67, M71,

F74, L75, V81, and I84) likely participate in the interaction (Fig. 4a). The polar residues of SutA (K60 and R64) and RNAP-β protrusion (S59 and E73) are in close proximity and likely form polar interaction with each other (Fig. 4b). Protein sequence alignment of SutA of non-redundant bacteria species reveals that the proposed interface residues are conserved (Fig. 4c). To validate the interface observed in the structure, we tested the interaction of wild-type or mutant SutA and RNAP-β protrusion using the YTH approach. The results showed that alanine substitution of the hydrophobic residues (L67, M71, F74, L75, V81, and I84) or polar residues (K60 and R64) of SutA at the interface substantially impaired SutA/β protrusion interactions (Fig. 4d). Moreover, bulky substitution of the two conserved glycine residues (G78 and G79) of SutA at the interface also substantially impaired the interaction (Fig. 4d), confirmed the structure model.

**SutA wedges open the RNAP-β lobe and widens the main cleft.** Structure superimposition of the SutA-RNAP-$\sigma^S$ (open lobe) and

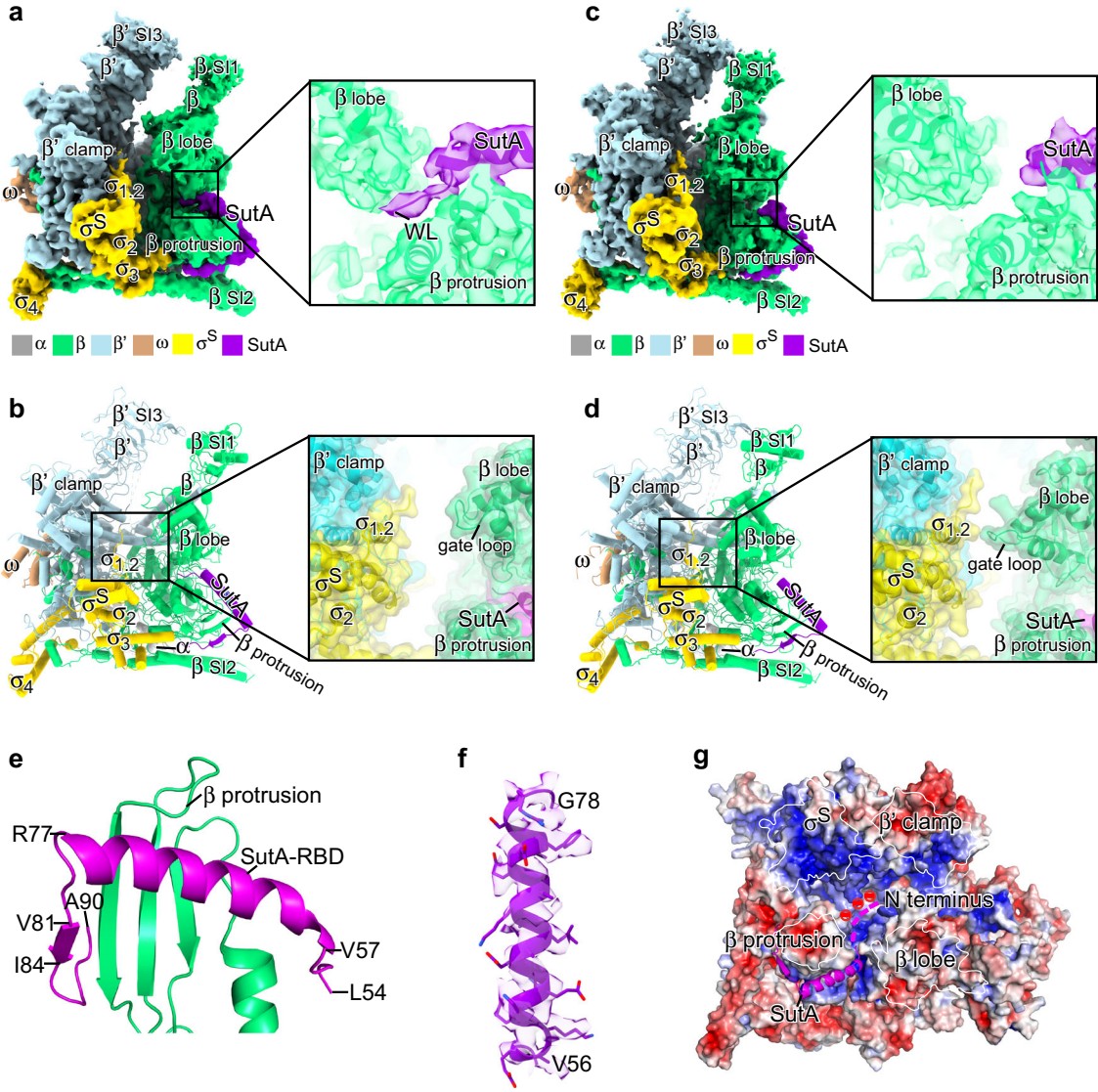

**Fig. 3 The cryo-EM structure of *Pae* SutA-RNAP-σ$^S$ complex. a** The cryo-EM map for SutA-RNAP-σ$^S$ complex (open lobe). RNAP subunits are colored as the color scheme. SutA is colored in purple. The insert shows the map signal of the wedge loop of SutA in the RNAP-β protrusion/lobe gap. **b** The structural model of SutA-RNAP-σ$^S$ complex (open lobe). The insert shows the surface presentation of the structure highlighting SutA wedge loop (SutA-WL) in RNAP-β protrusion/lobe gap and the open main cleft. **c** The cryo-EM map for SutA-RNAP-σ$^S$ complex (closed lobe). The insert shows no signal of the wedge loop of SutA in RNAP-β protrusion/lobe gap. **d** The structural model for SutA-RNAP-σ$^S$ complex (open lobe). The insert shows the surface presentation of the structure highlighting the disordered SutA-WL and the closed main cleft. **e** The interaction between SutA-RBD and RNAP-β protrusion. **f** The cryo-EM map for the middle helix of SutA-RBD. **g** The proposed route of the D/E-rich NTD (dashed purple line) entering the positively charged main cleft of RNAP. The electrostatic potential surface of RNAP was generated using APBS tools in Pymol.

RNAP-σ$^S$ holoenzyme complexes shows differences in RNAP-β lobe conformation and main cleft width. In the SutA-RNAP-σ$^S$ (open lobe) structure, the RNAP-β lobe domain swings ~10° away from RNAP-β' clamp resulting in ~5 Å increase in distance between RNAP-β' clamp and RNAP-β lobe compared with that of RNAP-σ$^S$ holoenzyme (Fig. 5a). The gate loop (RNAP-β residues 373-383; corresponding to RNAP-β residues 368-378 in *E. coli*), which serves as a rate-limiting gate for promoter loading in *E. coli* and *M. tuberculosis* RNAP, is opened along with the movement of β lobe (Fig. 5a). The main cleft of SutA-RNAP-σ$^S$ (open lobe) can accommodate the downstream dsDNA as shown by structural modelling (Fig. 5b). In sharp contrast, there exists substantial steric clashes between the modeled dsDNA and RNAP in the RNAP-σ$^S$ holoenzyme (Fig. 5c). The results suggested that SutA opens RNAP-β lobe and widens the main cleft to allow loading of downstream dsDNA into the main cleft.

As stated in the above section, we also trapped a closed conformation of RNAP-β lobe in SutA-RNAP-σ$^S$ (closed lobe). In the structure, the β lobe adopts a slightly more closed conformation than that of RNAP-σ$^S$ holoenzyme (Supplementary Fig. 5a), and the middle helix and its flanking C-terminal loop of SutA make essentially the same interactions with RNAP-β protrusion as in the structure of SutA-RNAP-σ$^S$ (open lobe) (Supplementary Fig. 5b). No signal was observed for the wedge loop and the N-terminal D/E-rich region of SutA in the structure of SutA-RNAP-σ$^S$ (closed lobe) (Fig. 3c, d and Supplementary Fig. 3b).

The conformational differences of the two SutA-bound RNAP-σ$^S$ structures are apparently attributed to the different interaction modes of SutA wedge loop. SutA wedge loop invades the RNAP-β lobe/protrusion gap in SutA-RNAP-σ$^S$ (open lobe) but is disordered in SutA-RNAP-σ$^S$ (closed lobe) (Fig. 3a, c and

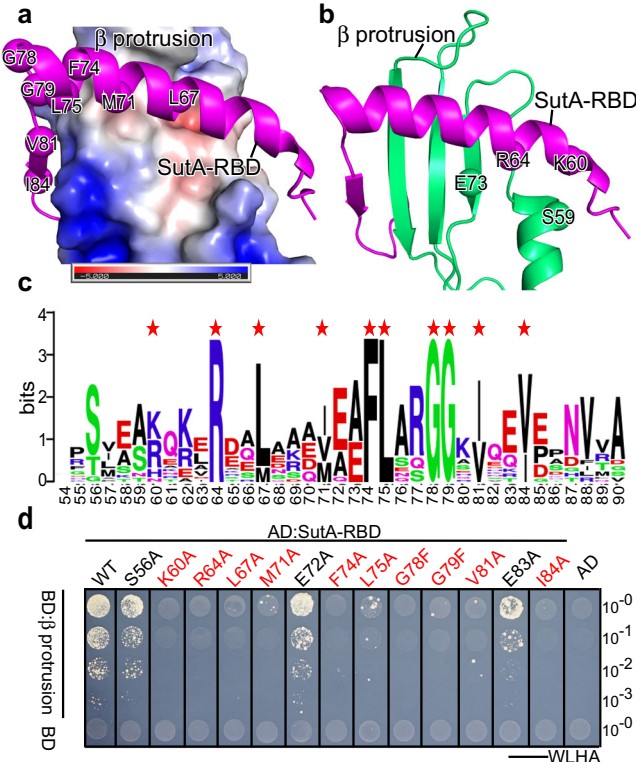

**Fig. 4 The detailed interactions between SutA-RBD and RNAP-β protrusion. a** The hydrophobic residues of SutA-RBD interact with a non-polar surface of RNAP-β protrusion. **b** The possible polar contacts made by nearby polar residues of SutA-RBD and RNAP-β protrusion. **c** Protein sequence alignment of SutA from 17 non-redundant bacterial species. Red asterisks indicate interface residues. **d** Alanine substitution of the interface residues disrupts interactions between SutA-RBD and RNAP-β protrusion. Source data are provided as a Source Data file.

Supplementary Fig. 3b). Structure superimposition shows that SutA wedge loop encounters steric hindrance with the β lobe domain in closed conformations of both RNAP-σ$^S$ (closed lobe) and RNAP-σ$^S$ holoenzyme (Fig. 5d, e and Supplementary Fig. 5c, d), suggesting that the invasion of SutA wedge loop induces opening of the β lobe. We further infer that the N-terminal D/E-rich SutA-NTD guides the wedge loop into the wedge position. Electrostatic surface presentation of RNAP shows that the main cleft of RNAP is highly positively charged (Fig. 3g), complementary to the natively charged SutA-NTD (23 D/E residues out of 55 residues in *Pae* SutA) (Fig. 1a). Therefore, SutA-NTD is likely captured in the main cleft and guides the wedge loop across the gap between RNAP-β protrusion and β lobe. Due to intrinsic flexibility of β lobe, the wedge loop slips in the bottom of the protrusion/lobe gap when the β lobe oscillates to its open conformation.

**SutA promotes RPo formation.** Because RNAP main cleft has to open to load the double-stranded promoter DNA during RPo formation[17,23–25], opening RNAP main cleft by SutA thereby is expected to facilitate promoter loading, and thus shift the equilibrium towards RPo formation. To test the hypothesis, we measured the kinetics of RPo formation by a stopped-flow fluorescence assay, in which the Cy3 fluorophore (attached to the +2 position of the non-template strand of promoter DNA) servers as a probe sensing local environment change and increases its fluorescence upon promoter unwinding[17,23–25]. The results showed that Cy3 fluorescence slowly reached a plateau, when RNAP-σ$^S$ holoenzyme alone was mixed with promoter DNA,

while SutA substantially increased the kinetics of RPo equilibration (Fig. 5f).

**σ$^A_{1.1}$ hinders the activation activity of SutA on RNAP-σ$^A$ holoenzyme.** SutA was initially identified as a transcription activator of RNAP-σ$^S$ holoenzyme that functions at slow-growth condition[1]. Later study suggested that SutA is also capable of activating RNAP-σ$^A$ holoenzyme (Fig. 1c), albeit with lower activity[2]. Previous reports showed that σ$^A_{1.1}$ occupies RNAP main cleft, makes substantial interactions with RNAP-β lobe, and restrains it in the closed conformation[11,26]. To study whether σ$^A_{1.1}$ hampers opening of RNAP-β lobe and thereby reduces the activity of SutA, we removed the σ$^A_{1.1}$ region and tested whether SutA activates transcription of RNAP-σ$^A$ (Δ1.1) holoenzyme. The in vitro transcription results showed that removal of domain σ$^A_{1.1}$ increased its basal transcription activity of σ$_{1.1}$, consistent with previous report showing σ$^A_{1.1}$ inhibits RPo formation (Fig. 5g)[13,18]. Moreover, SutA increases the transcription activity of RNAP-σ$^A$ (Δ1.1) holoenzyme by ~50%, close to the extent by which SutA activates RNAP-σ$^S$ holoenzyme (Fig. 5g). We propose deletion of σ$^A_{1.1}$ releases its restrain on RNAP-β lobe and the regained conformational flexibility allows SutA wedge loop to slip into the β protrusion/lobe gap and to lock the β lobe in an open conformation.

**The cryo-EM structure of *P. aeruginosa* SutA-σ$^S$-RPo.** To better understand the action of SutA upon RPo formation, we incubated SutA-RNAP-σ$^S$ with a duplex *rrn* promoter DNA (−49 to +21) for 20 s and immediately applied the reaction mixture on grids for cryo-EM data collection (Fig. 6a). Such strategy has been employed in previous studies to obtain structures of intermediate states during RPo formation[7,12,13]. However, we only obtained one major 3D class of single particles, from which a cryo-EM map at 5.8 Å resolution of SutA-bound RPo (SutA-σ$^S$-RPo) was reconstructed (Supplementary Table 2 and Supplementary Fig. 6). The map shows clear density for upstream promoter dsDNA (−35 to −12), 14 bp downstream dsDNA (+4 to +17), and the −10 element (−11 to −10) of non-template ssDNA of the transcription bubble (Fig. 6b, c and Supplementary Fig. 7d). The cryo-EM map also shows clear signals for SutA-RBD that interacts with RNAP-β protrusion as in the structure of SutA-RNAP-σ$^S$ (open lobe) (Fig. 6d), but no signal for SutA wedge loop, SutA-NTD, and SutA-CTD (Fig. 6b, c, and Supplementary Fig. 3b), indicating that SutA-RBD remains bound to RNAP upon RPo formation. Structure superimposition of SutA-σ$^S$-RPo and SutA-RNAP-σ$^S$ structures suggests that formation of RPo closes both the β' clamp and β lobe of RNAP (Fig. 6e).

**Discussion**
In this work, we have determined cryo-EM structures of *Pae* RNAP-σ$^S$ holoenzyme, *Pae* SutA-bound RNAP-σ$^S$ holoenzyme, and *Pae* SutA-σ$^S$-RPo. The structures show that SutA pinches RNAP-β protrusion, wedges β lobe, widens the main cleft, and facilitates promoter loading and unwinding. Our study supports a role of β lobe movement on promoter unwinding[16]. The main cleft is located between two pincers, RNAP-β' clamp and RNAP-β protrusion/lobe. It was long appreciated that RNAP-β protrusion/lobe is a rigid unit of RNAP, and that RNAP-β' clamp rotates to open or close the main cleft. Such assumption was made based on the fact that most reported bacterial RNAP structures show essentially the same closed conformation of β lobe/protrusion but different conformations of RNAP-β' clamp[27–33].

Until recently, molecular dynamics simulation of *T. aquaticus* RNAP revealed that RNAP-β lobe undergoes rapid oscillation between open and closed states, suggesting that RNAP-β lobe is

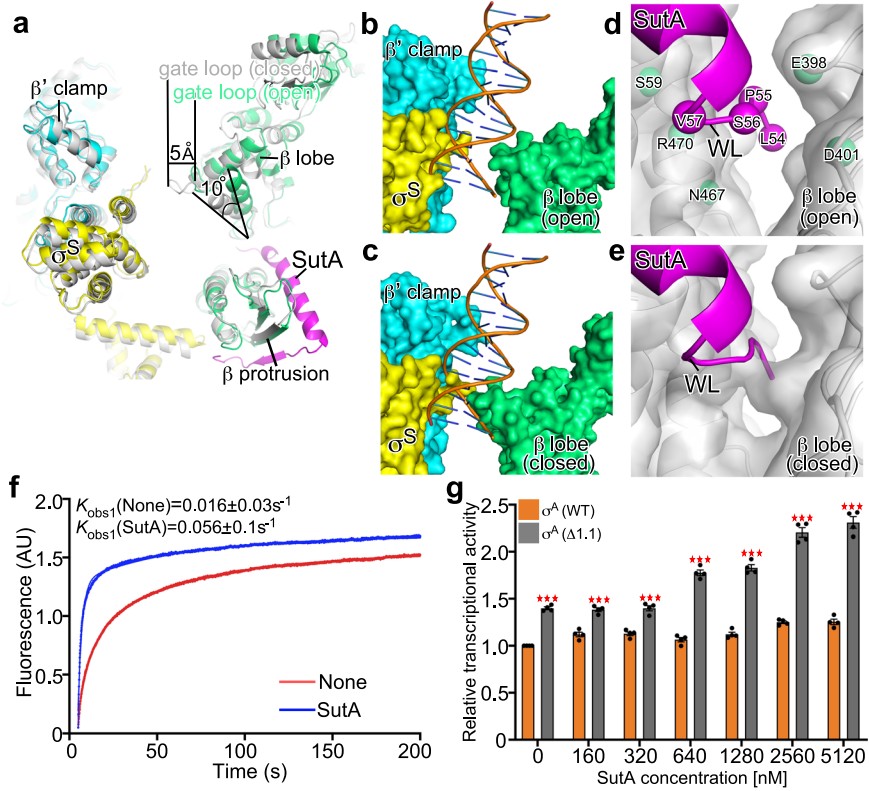

**Fig. 5 SutA widens the DNA cleft by opening β lobe. a** Structural superimposition between SutA-RNAP-$\sigma^S$ (open lobe; colored as above) and RNAP-$\sigma^S$ holoenzyme (gray) shows 10° opening of β gate loop and 5 Å increase in width of the RNAP main cleft. **b** Structure modeling shows that dsDNA is able to enter into the main cleft of SutA-RNAP-$\sigma^S$ (open lobe) with little steric hindrance. **c** The closed lobe in RNAP-$\sigma^S$ holoenzyme imposes a severe steric clash with dsDNA. **d** The wedge loop of SutA invades into the gap of RNAP-β protrusion/lobe in SutA-RNAP-$\sigma^S$ (open lobe). Spheres highlight the Cα atoms of residues that are likely involved in interactions of SutA-WL and RNAP-β protrusion/lobe. WL, wedge loop. **e** Structure modeling shows that the wedge loop of SutA encounters steric hindrance with the β lobe domain in closed conformation of RNAP-$\sigma^S$ holoenzyme. **f** The representative curves show SutA increases RPo formation equilibrium analyzed by a stopped-flow assay. **g** The results of fluorescence-based in vitro transcription assay show that SutA increases *rrn* promoter transcription by RNAP-$\sigma^A$ or RNAP-$\sigma^A$ (Δ1.1) holoenzymes. The data were presented as mean±S.E.M., $n = 4$. Two-sided Student's *t* test was performed. The individual data points were shown in black dots. *$p < 0.05$, **$p < 0.01$, ***$p < 0.001$, $n = 4$. Source data are provided as a Source Data file.

also intrinsically flexible[16]. A previously reported cryo-EM structure of corallopyronin A-bound *Mycobacterium tuberculosis* transcription initiation complex showed that the promoter DNA passed the β gate loop-$\sigma_{1.2}$ obstacle and occupied the main cleft in a partially melted form[12]. It is likely that the rotation of RNAP-β lobe accounts for opening of the main cleft to allow entry of the promoter DNA when the mobility of RNAP-β' clamp is inhibited by corallopyronin A. Here, our SutA-RNAP-$\sigma^S$ structures provide the direct structural evidence for the intrinsic flexibility of RNAP-β lobe, and further show that SutA restrains RNAP-β lobe in the open state to facilitate promoter loading and unwinding.

All evidence presented here lead to a model for structural mechanism of transcription activation by SutA (Fig. 7). The *rrn* promoter that contains a G/C-rich discriminator likely follows the "load-melt" pathway during promoter unwinding. SutA wedge loop opens the main cleft and facilitate the *rrn* promoter loading and unwinding. Loading of promoter DNA likely displaces SutA-NTD in the main cleft, disengages the wedge loop, and resumes the flexibility of lobe domain necessary for subsequent promoter unwinding steps.

Intriguingly, recent cryo-EM structures of TraR/DksA-bound transcription initiation complexes also reported the open state of β lobe (Supplementary Fig. 5e). In contrast, in those complexes, the β lobe is forced open by direct interaction between TraR/DksA and RNAP-β SI1, the non-conserved insertion extending

from RNAP-β lobe[7,13]. Together with these recent reports, our study pinpoints the importance of RNAP-β lobe for promoter unwinding and for regulation by diverse transcription regulators.

In summary, we report the structural mechanism for transcription activation of *Pae* SutA and propose a new means of transcription regulation. Our study presents structural evidence of intrinsic dynamic nature of RNAP-β lobe and an example that such the feature could be manipulated to affects kinetics of RPo formation by a transcription regulatory protein.

## Methods

**Plasmid construction**. Please see Supplementary Table 1 for the list of plasmids in this study.

**Proteins**. The *Pae* RNAP core enzyme was prepared from *E. coli* strain BL21(DE3) (Novo protein, Inc.) carrying pCOLA-*Pae rpoB-rpoC* and pACYC-*Pae rpoA-rpoZ*. Protein expression was induced at an $OD_{600}$ of 0.6 by 0.5 mM IPTG at 18 °C for overnight. Cells were harvested and resuspended in lysis Buffer A (40 mM Tris-HCl, pH 7.7, 200 mM NaCl, 5% glycerol, 2 mM EDTA, 2 mM DTT, 0.1 mM phenylmethylsulfonyl fluoride (PMSF) and protease inhibitor cocktail (Bioma-ke.cn. Inc.)) and lysed using an Avestin EmulsiFlex- C3 cell disrupter (Avestin, Inc.). The supernatant was precipitated with dropwise addition of 10% poly-ethylenimine (PEI) to a final concentration of 0.6%. The pellet was collected and RNAP was extracted with 100 mL of 10 mM Tris-HCl, pH 7.7, 5% glycerol, 1 M NaCl, 1 mM DTT, and 2 mM EDTA. RNAP was precipitated again by addition of ammonium sulfate (final concentration; 30 g/100 mL) and retrieved in 100 mL NTA-binding buffer (10 mM Tris-HCl, pH 7.7, 5% glycerol, 400 mM NaCl, 5 mM β-mercaptoethanol). The sample was applied onto a Ni-NTA column (Smart-

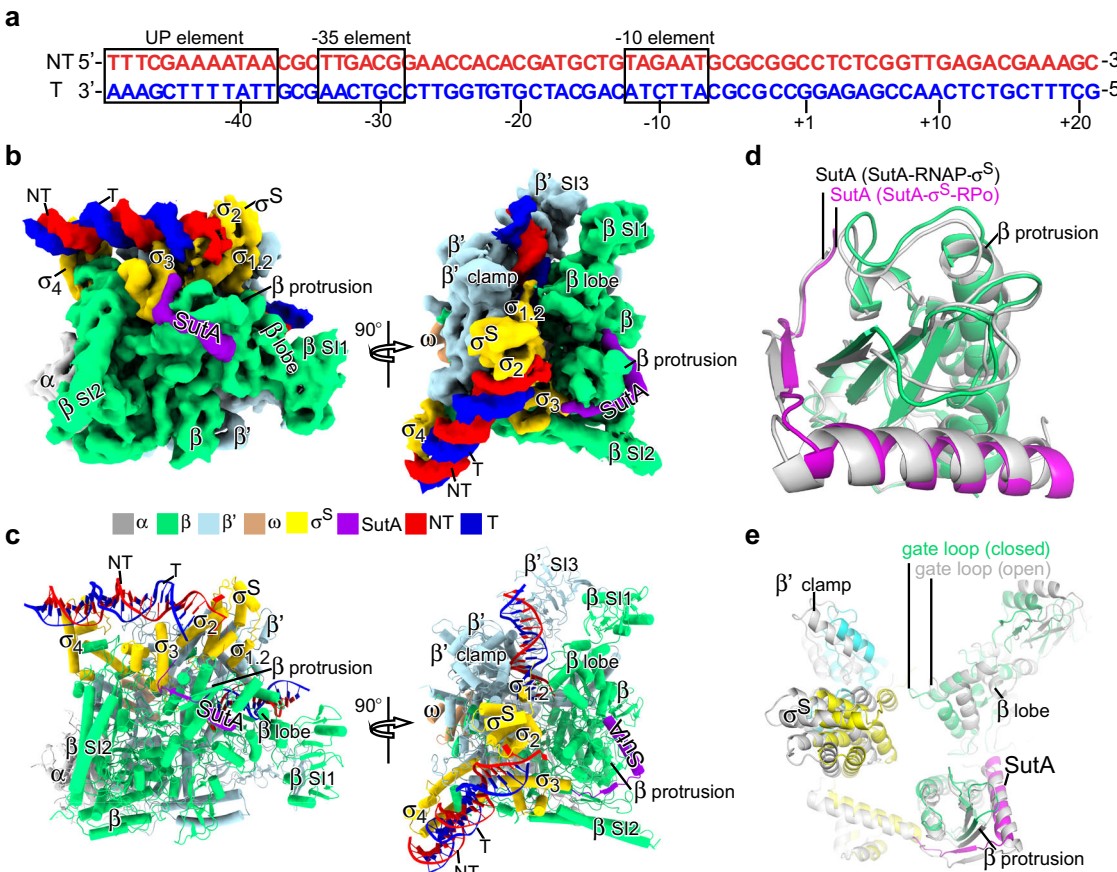

**Fig. 6 The cryo-EM structure of *Pae* SutA-σ^S-RPo. a** The promoter DNA used in structure determination of *Pae* SutA-σ^S-RPo. **b** The side and top view orientations of the cryo-EM map. RNAP subunits and SutA are colored as the color scheme. **c** The side and top view orientations of structure model of the SutA-σ^S-RPo. **d** SutA remains attached on and makes interaction with RNAP-β protrusion in the SutA-σ^S-RPo as it does in the structure of SutA-RNAP-σ^S. SutA-σ^S-RPo is colored as above and SutA-RNAP-σ^S (open lobe) is colored in gray. **e** Structural superimposition of SutA-RNAP-σ^S (open lobe) and SutA-σ^S-RPo shows the β lobe and β′ clamp is closed upon RPo formation. The SutA-σ^S-RPo is colored as above and SutA-RNAP-σ^S (open lobe) is colored in gray.

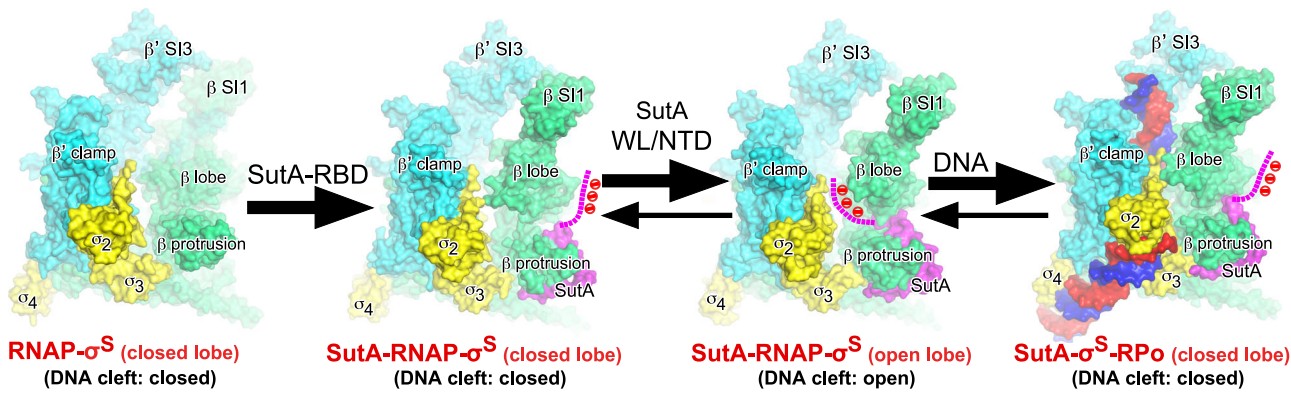

**Fig. 7 The proposed model for transcription activation by SutA.** RNAP-β lobe of RNAP holoenzyme is in rapid equilibrium between open and closed conformation but with preference on being closed. SutA anchors RNAP-β protrusion through its RBD and wedges open β lobe through its wedge loop to facilitate RPo formation.

Lifesciences Biotechnology), washed, and eluted with Ni-NTA buffer containing 300 mM imidazole. The eluted fractions were diluted and loaded onto a Mono Q column (MonoQ 10/100 GL, Cytiva) followed by a salt gradient of buffer A (10 mM Tris-HCl, pH 7.7, 200 mM NaCl, 5% (v/v) glycerol, 1 mM DTT, 0.1 mM EDTA) and buffer B (10 mM Tris-HCl, pH 7.7, 600 mM NaCl, 5% (v/v) glycerol, 1 mM DTT, 0.1 mM EDTA). The RNAP fractions were concentrated to 5 mg/mL and stored at −80 °C.

SutA was prepared from *E. coli* BL21(DE3) cells carrying pET-28a-TEV-SutA. The protein expression was induced with 0.4 mM IPTG at 18 °C for 20 h at OD_{600}

of 0.7. The cell pellet was lysed in lysis buffer B (50 mM Tris-HCl, pH 7.7, 500 mM NaCl, 5% (v/v) glycerol, 5 mM β-mercaptoethanol, 0.1 mM PMSF) using an Avestin EmulsiFlex-C3 cell disrupter. The supernatant was loaded on a 5 mL Ni-NTA column that was subsequently washed and eluted with lysis buffer B containing 300 mM imidazole. The eluted fractions were subjected to TEV protease, buffer-exchanged, and reloaded onto a Ni-NTA column to remove impurity. The fractions containing target proteins were concentrated to 4 mg/mL, and stored at −80 °C. *Pae* σ^S, σ^A and their derivatives were prepared by the same procedure.

**Nucleic-acid scaffolds for cryo-EM structure determination.** The nontemplate-strand ssDNA (1.1 mM final concentration; Sangon Biotech), template-strand ssDNA (1.2 mM final concentration; Sangon Biotech) in annealing buffer (5 mM Tris-HCl, pH 8.0, 200 mM NaCl and 10 mM MgCl$_2$) were heated for 5 min at 95 °C and cooled to 22 °C in 2 °C steps with 30 s per step using a thermal cycler.

**_Pae_ RNAP-σ$^S$-holoenzyme.** _Pae_ RNAP core enzyme and σ$^S$ were incubated in a 1:3 molar ratio for overnight at 4 °C and loaded onto a Superdex 200 Increase 10/300 GL column (Cytiva) equilibrated in 10 mM HEPES, pH 7.5, 100 mM KCl, 5 mM MgCl$_2$, 2 mM DTT. Fractions containing _Pae_ RNAP-σ$^S$-holoenzyme were collected and concentrated to 15 mg/mL.

**_Pae_ SutA-RNAP-σ$^S$.** _Pae_ RNAP core enzyme, σ$^S$ and SutA were incubated in a 1:3:3 molar ratio for overnight at 4 °C and loaded onto a Superdex 200 Increase 10/300 GL column (Cytiva) equilibrated in 10 mM HEPES, pH 7.5, 100 mM KCl, 5 mM MgCl$_2$, 2 mM DTT. Fractions containing _Pae_ SutA-RNAP-σ$^S$ were collected and concentrated to 12 mg/mL.

**Cryo-EM structure determination of _Pae_ RNAP-σ$^S$-holoenzyme.** The _Pae_ RNAP-σ$^S$ holoenzyme complex was freshly prepared as described above and mixed with CHAPSO (Hampton Research, Inc.; final concentration 8 mM) prior to grid preparation. About 3 μL sample was applied onto the glow-discharged C-flat CF-1.2/1.3 400 mesh holey carbon grids (Electron Microscopy Sciences) in the chamber of a Vitrobot Mark IV (FEI; 95% chamber humidity at 10 °C). The grids were subsequently plunge-frozen in liquid ethane.

The micrographs were collected using Serial EM on a 300 keV Titan Krios (FEI) equipped with a Gatan K2 Summit direct electron detector (pixel size 1.307 Å/pixel). A total of 1080 images were recorded using the counting mode (exposure, 10 s per 40-frame movie; dose rate, 10.0 electrons/pixel/s; defocus, −1.5 to −2.5 μm). Frames of individual movies were aligned using MotionCor2[34] and CTF estimations were performed using CTFFIND4[35]. 2D classes based on ~1000 particles were served as templates for particle picking. The resulting 358,323 particles were subjected to 2D classification in RELION 3.0 with a E. coli RNAP holoenzyme structure (50 Å low-pass-filtered) as the starting reference model for 3D classification (N = 4). One 3D class with distinct shape of RNAP containing 165,721 particles were subjected to 3D classification (N = 3) again, the resulting one class contains 123,317 particles was selected and subjected to 2D classification for generating templates for the second round of particle auto-picking. A total of 489,361 particles were auto-picked by using the 2D references. The resulting particles were subjected to 2D classification in RELION 3.0 by specifying 100 classes. A total of 428,544 particles were selected and subjected to 3D classification using a 40 Å low-pass-filtered cryo-EM structure of the previous 3D classification. One 3D class with distinct shape of RNAP containing 145,929 particles were subjected to 3D classification (N = 3) again, two classes were combined and used for 3D auto-refinement, CTF-refinement and Bayesian polishing, resulting a 4.69 Å map. To resolve RNAP heterogeneity around β lobe, a soft mask that excluded the RNAP-β protrusion and RNAP-β lobe regions was generated in Chimera and RELION 3.0. The mask was used to make a subtracted particle stack in RELION 3.0. The subtracted particles were applied for masked 3D classification (N = 3, without alignment) resulting in one major 3D class. The 105,629 particles in the 3D class were reverted and processed through iteratively 3D auto-refinement, CTF-refinement, Bayesian polishing and post-processing in RELION 3.0. Gold-standard Fourier-shell-correlation analysis (FSC) indicated nominal resolutions of 4.05 Å for the final mode of RNAP-σ$^S$ holoenzyme. E. coli RNAP-σ$^A$ holoenzyme model extracted from E. coli RNAP-σ$^A$ RPo (PDB: 4YLN)[36] was manually fit into the cryo-EM map using Chimera and refined using Coot and phenix.

**Cryo-EM structure determination of _Pae_ SutA-RNAP-σ$^S$.** The SutA-RNAP-σ$^S$ complex was freshly prepared as described above and mixed with CHAPSO (Hampton Research, Inc.; final concentration 8 mM) prior to grid preparation.

About 3 μL mixture was applied onto the glow-discharged Quantifoil R1.2/1.3 300 mesh holey carbon grids (Quantifoil) in the chamber of a Vitrobot Mark IV (FEI; 100% chamber humidity at 22 °C), and the grids were subsequently plunge-frozen in liquid ethane.

The sample was prepared essentially as above. The micrographs were collected using EPU in the super-resolution counting mode on a 300 keV Titan Krios (FEI) equipped with a Gatan K3 Summit direct electron detector (pixel size 1.10 Å/pixel). A total of 2,805 images were recorded using the counting mode (exposure, 2.67 s per 40-frame movie; dose rate, 22.5 electrons/pixel/s; defocus, −1.2 to −2.2 μm). The data were processed essentially as above. To resolve RNAP heterogeneity around SutA, the same soft mask that excluded the RNAP-β protrusion and RNAP-β lobe regions was generated in Chimera and RELION 3.0. The mask was used to make a subtracted particle stack in RELION 3.0. The subtracted particles were applied for masked 3D classification (N = 2, without alignment) resulting in two 3D classes with distinct conformations of RNAP-β lobe. The single particles of two 3D classes were separately reverted and processed through 3D auto-refinement and post-processing in RELION 3.0. Gold-standard Fourier-shell-correlation analysis (FSC) indicated nominal resolutions of 3.13 Å and 3.86 Å for the final modes of SutA-RNAP-σ$^S$ (open lobe) and SutA-RNAP-σ$^S$ (closed lobe),

respectively. For building RNAP-β lobe model of SutA-RNAP-σ$^S$ (open lobe), we imported the single particles of SutA-RNAP-σ$^S$ (open lobe) into cryoSPARC v2.15. The 138,109 particles were subjected to non-uniform refinement, resulting in a 3.86 Å map. The soft mask that excluded the RNAP-β protrusion /lobe-SI1 regions was generated in Chimera. The mask was used to make a subtracted particle stack and then the subtracted particles were applied for local refinement and sharpening in cryoSPARC v2.15. Gold-standard Fourier-shell-correlation analysis (FSC) indicated nominal resolutions of 4.77 Å. The structure model of _Pae_ RNAP-β protrusion/lobe was refined using phenix. The _Pae_ RNAP-σ$^S$ was manually fit into the cryo-EM maps using Chimera and refined using Coot and phenix. The middle helix (residues 57-76) of SutA was first fit into the crescent-shaped density map. The residue register of the helix was determined by density map of bulky residue sidechains and the resulting model agrees to previous BPA cross-linking data[2]. The structural model was subsequently refined in Phenix.

**Cryo-EM structure determination of _Pae_ SutA-σ$^S$-RPo.** _Pae_ SutA-σ$^S$-RPo (25 μM) was mixed with _rrn_ full duplex promoter DNA (38 μM) with a molar ratio of 1:1.5 for 20 s, and subsequently supplemented with CHAPSO (Hampton Research, Inc., final concentration 8 mM). About 3 μL mixture was applied onto the glow-discharged C-flat CF-1.2/1.3 400 mesh holey carbon grids (Electron Microscopy Sciences) in the chamber of a Vitrobot Mark IV (FEI; 95% chamber humidity at 10 °C), and the grids were subsequently plunge-frozen in liquid ethane. The total estimated reaction time is ~20 s.

The micrographs were collected using Serial EM on a 300 keV Titan Krios (FEI) equipped with a Gatan K2 Summit direct electron detector (pixel size 1.0 Å/pixel). A total of 2005 images were recorded using the counting mode (exposure, 7.6 s per 38-frame movie; dose rate, 8.0 electrons/pixel/s; defocus, −1.2 to −2.2 μm). The data were processed in a similar procedure as described above except auto-picking particles were subjected multiple rounds of 2D classification to remove junk particles. The 3D class containing 112,959 particles were subjected to 3D auto-refinement, CTF-refinement, Bayesian polishing, and post-processing steps in RELION 3.0. Gold-standard Fourier-shell-correlation analysis (FSC) indicated a nominal resolution of 5.77 Å. _Pae_ SutA-RNAP-σ$^S$ holoenzyme and promoter DNA from a E. coli RPo (PDB: 7KHB)[13] were manually fit into the cryo-EM map using Chimera. Rigid body and real-space refinement were performed in Coot and Phenix.

**Yeast two-hybrid assay.** The pGADT7-SutA (or pGADT7-SutA derivatives) and pGBKT7-β protrusion (or pGBKT7-β protrusion derivatives) were transformed into the yeast strain AH109, respectively. The yeast AH109 cells bearing paired plasmids were mated and spotted on agar plates of a stringent selective medium without adenine, histidine, tryptophan, and leucine, or a selective medium without leucine and tryptophan. The growth of yeast colonies were recorded after 5 days.

**Fluorescence labeling of σ$_{1.1}$.** _Pae_ σ$_{1.1}$$^{(A2C)}$ (σ$_{1.1}$ $^{(A2C)}$ derivative bearing an Ala to Cys substitution at residue position 2) was labeled with fluorescein at residue Cys[2]. The reaction mixture (2 mL) containing σ$_{1.1}$$^{(A2C)}$ (2 mM) and Fluorescein-5-Maleimide (20 mM Thermo Scientific, Inc.) in PBS, was incubated for 2 h at room temperature. The reaction was terminated by addition of DTT (1 mM; final concentration) and the labeled protein was purified by a 5 mL PD-10 desalting column (Biorad, Inc.). The fractions containing labeled protein were pooled and concentrated to 0.5 mg/mL. The _Pae_ σ$_{A1.1}$$^{(S2C)}$ was labelled in a similar procedure as described above.

**Fluorescence polarization assay.** The reaction mixtures (100 μL) for measuring the binding affinity of σ$_{1.1}$ and RNAP include F5M-σ$_{A1.1}$$^{(S2C)}$ or F5M-σ$_{S1.1}$$^{(A2C)}$ (final concentration: 2 nM), _Pae_ RNAP core enzyme (finial concentrations: 0 nM,16 nM, 32 nM, 64 nM, 128 nM, 256 nM, 512 nM,1024 nM, 2048 nM, and 4096 nM) in FP buffer (10 mM Tris-HCl, pH 7.7, 100 mM NaCl,1 mM DTT, 1% glycerol, and 0.025% Tween-20). The mixtures were incubated in a 96-well plate (Corning, Inc) for 10 min at room temperature. The fluorescence polarization (FP) signals were collected using SparkControl software on a plate reader (SPARK, TECAN Inc.) equipped with excitation filter of 485/20 nm and emission filter of 520/20 nm. The data were plotted in Prism v.8.4.0 (GraphPad Software) and the dissociation constant Kd values were estimated by fitting data to the following equation,

$$F = B[S]/(Kd + [S]) + F_0$$

Where F is the FP signal at a given concentration of RNAP, $F_0$ is the FP signal in the absence of RNAP, [S] is the concentration of RNAP or, and B is an unconstrained constant.

**Stopped-flow assay.** To monitor promoter unwinding by _Pae_ RNAP-σ$^S$ holoenzyme, 60 μL _Pae_ RNAP-σ$^S$ holoenzyme (final concentration: 40 nM) or pre-assembled _Pae_ SutA-RNAP-σ$^S$ holoenzyme (final concentration: 40 nM) and 60 μL Cy3-λP$_R$ promoter DNA (final concentration: 1 nM; prepared as in[37]) in buffer (10 mM Tris-HCl, pH 7.5, 200 mM NaCl, 10 mM MgCl$_2$, 1 mM DTT) were rapidly mixed and the change of Cy3 fluorescence was monitored in real time by a

stopped-flow instrument (SX20, Applied Photophysics Ltd, UK) equipped with a excitation filter (515/9.3 nm) and a long-pass emission filter (570 nm). The data were collected using Pro-Data SX software and were plotted in Prism (GraphPad, Inc.) and the observed rates $k_{obs,1}$ and $k_{obs,2}$ were estimated by fitting the data with following equation,

$$F = F_0 + a(1 - e^{-k_1 t}) + b(1 - e^{-k_2 t})$$

Where F is the fluorescence value at a giving time point, $t$ is the reaction time, $F_0$ is the fluorescence value at point of $t = 0$, $k_1$ and $k_2$ are the observed rates $k_{obs,1}$ and $k_{obs,2}$, a and b are unrestrained constants.

**Fluorescence-detected in vitro transcription assay**. The experiment was performed as in[38]. Briefly, the reaction mixtures (20 μL) contain *Pae* RNAP holoenzyme (final concentration: 40 nM), *rrn* promoter DNA (final concentration: 40 nM; sequence: 5′-cggcgcaagcggttgagtagaaaagaaaattttcgaaaataacgcTTGACGgaacgaga ggttgctgTAGAATgcgcggcctcggttgagacgaaagccttgaccaactgctctttaacaagtcgaatcagg cacgtacgaaggaaggattggtatgtggtatattcgtacgtgccggcctgctggtaatcgcaggccttttttatttaaggg cagcttggcgtaatcatggtcatagctgtttcctgctgtg-3′; the sequence of Mango riboswitch is underscored and the −35/−10 elements are in upper case), SutA (finial concentration: 0 nM, 160 nM, 320 nM, 640 nM, 1280 nM, 2560 nM, 5120 nM) in the reaction buffer (50 mM Tris-HCl pH 7.9, 100 mM KCl, 10 mM MgCl₂, 1 mM DTT, 5% glycerol and 0.01% Tween-20) were incubated at 37 °C for 10 min. The reactions were initiated by addition of NTP mix (final concentration: 0.1 mM each) and TOl-3PEG-Biotin (final concentration: 0.5 μM) followed by 30 min incubation at 37 °C. The fluorescence signals were collected using SparkControl software on a plate reader (SPARK, TECAN) at an excitation wavelength of 510/10 nm and an emission wavelength of 550/10 nm.

**Measurement of RNAP-β lobe movement**. The rotation angle of β lobe and the main-cleft width change of the two states (open and closed β lobe) was measured as in[16]. Briefly, the rotation angle was defined as the angle of Cα atoms of three residues (βQ364 of open lobe, βN455 in closed lobe, and βQ364 in closed lobe; the position of βN455 is unchanged in both structure). The width change of the main cleft in the two states (open and closed β lobe) was defined as change of the distance between Cα atoms of βK383 and σL67.

**Structural modeling**. We modelled 16-base pair double stranded B-DNA in the main cleft between the β' clamp and β lobe of the SutA-RNAP-σ^S (open lobe) using the PyMOL Molecular Graphics System (Version 2.0 Schrödinger, LLC) and checked steric clash between the modeled DNA and protein on MDWeb webserver.

**Reporting summary**. Further information on research design is available in the Nature Research Reporting Summary linked to this article.

## Data availability

The data that support this study are available from the corresponding authors upon reasonable request. The cryo-EM density maps have been deposited in the Electron Microscopy Data Bank (EMDB) under accession numbers EMD-31948 (*Pae* RNAP-σ^S holoenzyme), EMD-33271 (*Pae* SutA-RNAP-σ^S open lobe), EMD-33272 (*Pae* SutA-RNAP-σ^S closed lobe), EMD-31403 (*Pae* SutA-σ^S-RPo). The coordinates have been deposited in the RCSB Protein Data Bank (PDB) under accession codes 7VF9 [https://doi.org/10.2210/pdb7VF9/pdb] (*Pae* RNAP-σ^S holoenzyme), 7XL3 [https://doi.org/10.2210/pdb7XL3/pdb] (*Pae* SutA-RNAP-σ^S open lobe), 7XL4 [https://doi.org/10.2210/pdb7XL4/pdb] (*Pae* SutA-RNAP-σ^S closed lobe), 7F0R [https://doi.org/10.2210/pdb7F0R/pdb] (*Pae* SutA-σ^S-RPo), 4YLN [https://doi.org/10.2210/pdb4YLN/pdb] (*E. coli* RNAP-σ^A transcription initiation complex). The source data underlying Figs. 1b, c, 2c, 4d, 5f, g are provided as a Source Data file. Source data are provided with this paper.

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

## Acknowledgements

This work was supported by the National Key Research and Development Program of China (2018YFA0900701 to Y.Z.), the CAS Strategic Priority Research Program (XDB29020000 to Y.Z.), the Shanghai Science and technology innovation program (19JC1415900 to Y.Z.), and the National Natural Science Foundation of China grant (31670067 and 31822001 to Y.Z.), the National Natural Science Foundation of China (31970040 to Y.F.) and Natural Science Foundation of Zhejiang Province (LR21C010002 to Y.F.). We thank Dr. Ilona Christy Unarta and Prof. Xuhui Huang in the Hong Kong University of Science and Technology for assisting in analyzing the dsDNA accessibility in SutA-RNAP-σ$^S$ and RNAP-σ$^S$ structures. We thank Dr. Liangliang Kong, Dr. Fangfang Wang, Dr. Guangyi Li, and Dr. Jialin Duan at the Electron Microscopy System of the National Facility for Protein Science in Shanghai (NFPS), Dr. Shenghai Chang at the cryo-EM center of Zhejiang University, for assistance with data collection.

## Author contributions

D.H., and L.Y. collected cryo-EM data and determined structures. D.H. and X.W. performed stopped-flow experiments. D.H. performed FP and in vitro transcription assay. D.H., and L.Y. performed structural determination. C.F., J.S., A.W., and Y.F. assisted in data collection. W.M. and Z.Y. assisted in protein preparation. Y.F. provided cryo-EM facility access and assisted in cryo-EM data collection and processing. Y.Z. designed experiments, analyzed data, and wrote the manuscript.

## Competing interests

The authors declare no competing interests.
