## [Peer Review File · Nature Communications]

Pseudomonas aeruginosa SutA wedges RNAP lobe domain open to facilitate promoter DNA unwindingReviewers' Comments:

Reviewer #1:

Remarks to the Author:

This manuscript describes three transcription complex structures of *Pseudomonas aeruginosa* at modest resolution (3.9-5.8 Å): the holoenzyme containing RNAP and sigmaS, the stress response sigma, the holoenzyme in complex with SutA, a transcription factor that binds to RNAP and is induced under anaerobic conditions, as well as the SutA bound open promoter complex. The resolution is insufficient to precisely model SutA. Instead the authors used homology model and mutagenesis to identify potential interaction sites between SutA and RNAP. Interestingly, the authors show that SutA-holoenzyme complex exhibits both open and closed β -lobe in RNAP while in the holoenzyme structure, only closed β -lobe conformation is observed. Further SutA binding seems to relocate $\sigma 1.1$ of sigmaS, which is shown to occupy the downstream DNA binding channel. Based on these structural observations, the authors suggest that SutA opens up β -lobe while helping dislodging $\sigma 1.1$, probably through its acidic N-terminal domain, the latter is supported by biochemical data. Further, the authors also present a SutA bound open promoter complex structure. Opening up β -lobe and dislodging $\sigma 1.1$ would promote open promoter complex formation. Based on these observations, the authors propose SutA activates sigmaS-dependent transcription through opening up the β -lobe and dislodging $\sigma 1.1$. Intriguingly SutA does not activate sigmaA dependent transcription. However when $\sigma 1.1$ of sigma is deleted, adding saturating amount of SutA does increase in vitro transcription, the authors thus suggest $\sigma 1.1$ is the determinants for SutA.

While the work overall is of high quality, the conclusions are not fully supported by data presented here. I will list my comments and concerns below:

- The structural changes observed are in the absence of DNA, hence it is unclear if indeed these changes will be preserved in the promoter DNA complex. Unfortunately incubating with duplex DNA leads to open complex formation. And the open complex conformation is the same as those in the absence of SutA, thus unable to delineate the contributions of SutA in open complex formation.
- Given the extensive interface observed between sigmaS and RNAP, I am surprised that the authors are unable to measure the interactions between sigmaS and RNAP using fluorescent polarization experiment. The authors argue that this shows the interactions between sigmaS and RNAP is weak. This is in contrast with their cryoEM structure. Further, these negative results raise questions about the sensitivity of the FP experiments and thus the reliability of the FP experiments presented in the manuscript.
- The authors suggest that the holoenzyme exists in an equilibrium of open and closed β -lobes and SutA binding opens up the β -lobe, widening the downstream DNA binding channel, thus helping with open complex formation. While this is totally plausible, the data here do not support this conclusion as the holoenzyme only shows closed β -lobe conformation (did not show an equilibrium of open and closed β -lobe conformations). In the presence of SutA, both the open and closed β -lobe conformations are observed. Can the authors show what differences are there in these two conformations and how SutA binding would promote open β -lobe? Further, the authors show that in the RPo, β -lobe is closed while SutA retains the same interactions with RNAP-sigmaS. Again it is unclear how SutA promotes β -lobe opening, thus it is unclear how it promotes open complex formation.
- The authors further suggest that the N-terminal acidic regions might compete with $\sigma 1.1$ for downstream DNA binding channel, explaining the absence of $\sigma 1.1$ density in the SutA bound complex. If this is the case, presumably some complexes will have $\sigma 1.1$ still in place while others have SutA N-terminal domain occupying this space. Have the authors looked carefully to see if there are two different complexes (classes) in their datasets, one with $\sigma 1.1$ in place and the other one with SutA in place?
- The authors showed that transcription by RNAP-sigmaS and RNAP-sigmaA ($\Delta\sigma 1.1$) could be stimulated by SutA (at very high concentration) while RNAP- σA can't. The authors thus suggest that $\sigma 1.1$ region is the determination of SutA specificity on RNAP-sigmaS. Exactly how this is achieved is unclear. It would be interesting to elaborate how this is the case based on the structures. Further, the

authors propose that SutA N terminal domain competes out sigmaS, but not sigmaA as sigmaA σ 1.1 binds tighter to the downstream DNA channel. Given that σ 1.1 has an inhibitory effect on open complex formation, I am surprised that when σ 1.1 is deleted (Figure 5D), in the absence of SutA, the transcription effects are not stimulated. Further, if N-terminal domain of SutA competes out sigmaS σ 1.1 in downstream DNA binding channel, SutA should bind to downstream DNA channel more tightly than sigmaS, hence would impose higher inhibition for downstream DNA to enter the cleft, thus should be more inhibitory to open complex formation. Taken these data together, it is not clear that SutA competing with σ 1.1 in occupying the downstream DNA channel is one of the mechanisms that promotes transcription.

Overall the data presented here reveal interesting properties of SutA but do not convincingly address the molecular mechanisms of SutA in stimulating sigmaS dependent transcription. It would be important for the authors to resolve the functionality of the N-terminus of SutA and to understand how SutA binding induces the conformational changes observed (open β -lobe and dislodging of σ 1.1) in order to reveal the mechanisms of how SutA stimulates transcription. It is possible that with better resolution, the authors might be able to resolve SutA N-terminus in their structures, thus correlating precise SutA binding with the conformation of RNAP-sigmaS. What would the structure look like when SutA N-terminal domain is deleted ?

Reviewer #2:

Remarks to the Author:

He, You et al. have provided structures of the Pseudomonas transcription factor, SutA, with SigS-RNAP. While the observation and conclusions are potentially interesting, there are some red flags on the cryo-EM statistics and analyses. The main new observations are that SutA displaces region 1.1 of SigS more effectively than SigA, explaining its more potent effect on SigS. The second exciting observation is that SutA interacts with the protrusion (previously shown biochemically but shown here structurally), which might lead to a shift in equilibrium to the lobe of RNAP towards the open state. What's unclear and the authors should state so or postulate a mechanism to how that interaction causes the lobe to be more open. Also, it's a paradox that SutA removes region 1.1 of SigS but also then occupies the same channel (seemingly more stably than region 1.1), which would also compete with DNA placement. Can the authors address this? However, this paper's main detraction is that particle distribution analyses appear to show severe particle orientations in all structures here, with most views unrepresented. Therefore, the authors must present this data using a 3D-FSC analysis. In addition, the authors need meshed density maps and local resolution maps highlighting the features of SigS (region 1) and all of SutA in all structures, so the readers get an idea of the resolution and interpretable densities described in the manuscript. Supplementary Fig 5D is very strange- the local resolution goes from 7 Å to 4 Å without anything in between? Are the authors sure they did this analysis correctly? Minor comments are listed below:

- Pg3 ln 71: this is incorrect, according to the cited paper: '(SutA) activates transcription from the *rrn* promoter by both the housekeeping sigma factor holoenzyme (E σ 70) and the stress sigma factor holoenzyme (E σ S) in vitro, but has a greater impact on E σ S.'
- Pg4 ln 106: Specificity for SigS is questionable; also binds SigA (resulting in *rrn* upregulation)
- C-term truncation of SutA appears to incompletely suppress the impact of SutA on transcription in Fig1C assay; this should be mentioned
- Raw fluorescence values or images of the mango riboswitch in operation would help to illustrate the effect in Fig 1C
- Claiming that the visual absence of sigma 1.1 in SutA+Holo structure indicates 'incompatibility between Sigma 1.1 and SutA' is not necessarily supported. Sigma 1.1 might just be freer to move around in the SutA+Holo structure due to overall clamp-lobe orientation- should be stated as an alternative hypothesis. Also, one would need to have the two maps with the same levels and contour and a close-up view. Pymol can weigh the maps similarly. Also, if SutA can outcompete SigS1.1, why

don't we see it in the map (we see SigS 1.1). The authors should address that.

- The fluorescence anisotropy assay might require some troubleshooting to see SigS 1.1 binding, as the cryo-em maps suggest it must bind RNAP to some detectable extent
- 4096 nM is a lot of SutA; I'd be cautious in using this to determine biologically relevant binding tendencies of SutA to holo
- 'the structural and biochemical evidence suggests that SutA-RBD anchors to RNAP-b protrusion and SutA-NTD invades the main cleft to open RNAP-b lobe and release SigS1.1 from the main cleft.'
 - o I would caveat this claim; SutA-NTD is nowhere to be seen in the maps.
- The idea that SutA-NTD invades the cleft, widens the clamp-lobe spacing, and displaces sigA 1.1 to thereby facilitate DNA loading is problematic
 - o It is claimed that SigS 1.1 binds so weakly that fluorescence polarization assays can't detect the binding, and yet SutA-NTD, which successfully outcompetes and displaces the far more challenging SigA 1.1, is supposed to be an easier obstacle for DNA loading to overcome than SigS 1.1? This is inconsistent with the proposed model.
- Description of the stopped-flow assay and how it illustrates RPO formation wasn't very clear
- Fig 5D, are these the same assays as 1C- if so, please state in legend? Also, it would be good to illustrate this effect with SigS 1.1 as well (full length and 1.1 truncations)
 - o Conclusions from this assay are inconsistent with the paper's earlier remarks (line 71)
 - o Loss of this effect for SigS and SigA 1.1 regions where truncated SutA (missing NTD) is used would help to corroborate the paper's model
- Line 263, the word 'observation' is inappropriate here as there is no SutA-NTD visible in these maps (as far as this figure shows)

Reply to reviewers' comments:

Reviewer #1 (Remarks to the Author):

This manuscript describes three transcription complex structures of *Pseudomonas aeruginosa* at modest resolution (3.9-5.8 Å): the holoenzyme containing RNAP and σ^S , the stress response sigma, the holoenzyme in complex with SutA, a transcription factor that binds to RNAP and is induced under anaerobic conditions, as well as the SutA bound open promoter complex. The resolution is insufficient to precisely model SutA. Instead the authors used homology model and mutagenesis to identify potential interaction sites between SutA and RNAP. Interestingly, the authors show that SutA-holoenzyme complex exhibits both open and closed β -lobe in RNAP while in the holoenzyme structure, only closed β -lobe conformation is observed. Further SutA binding seems to relocate $\sigma^{1.1}$ of σ^S , which is shown to occupy the downstream DNA binding channel. Based on these structural observations, the authors suggest that SutA opens up β -lobe while helping dislodging $\sigma^{1.1}$, probably through its acidic N-terminal domain, the latter is supported by biochemical data. Further, the authors also present a SutA bound open promoter complex structure. Opening up β -lobe and dislodging $\sigma^{1.1}$ would promote open promoter complex formation. Based on these observations, the authors propose SutA activates σ^S -dependent transcription through opening up the β -lobe and dislodging $\sigma^{1.1}$. Intriguingly SutA does not activate σ^A dependent transcription. However when $\sigma^{1.1}$ of sigma is deleted, adding saturating amount of SutA does increase in vitro transcription, the authors thus suggest $\sigma^{1.1}$ is the determinants for SutA. While the work overall is of high quality, the conclusions are not fully supported by data presented here. I will list my comments and concerns below:

Reply: We thank the review for the thoughtful comments and suggestions. In the revised manuscript, we have recollected a dataset for SutA-RNAP- σ^S and substantially improved map resolution and model quality of two structures. The resolution of the SutA-RNAP- σ^S (open lobe) has been improved from 3.9 Å to 3.1 Å and the resolution of the SutA-RNAP- σ^S (closed lobe) has been improved from 4.1 Å to 3.9 Å. The improved map allows us to confidently model the RNA-binding domain of SutA, and more importantly, the new map with better resolution allows us to resolve a key loop (wedge loop) of SutA that wedges RNAP β lobe open in the structure of SutA-RNAP- σ^S (open lobe). The wedge loop is incompatible with a closed lobe and therefore is disordered in the structure of SutA-RNAP- σ^S (closed lobe). Based on the new structures, we revised SutA working model (Fig. 7). We propose that SutA pinches RNAP- β protrusion with its RNAP-binding domain and wedges open RNAP- β lobe domain to facilitate promoter unwinding with its wedge loop and N-terminal D/E-rich region.

- 1. The structural changes observed are in the absence of DNA, hence it is unclear if indeed these changes will be preserved in the promoter DNA complex. Unfortunately incubating with duplex DNA leads to open complex formation. And the open complex conformation is the same as those in the absence of SutA, thus unable to delineate the contributions of SutA in open complex formation.**

Reply: Our new structure of SutA-RNAP- σ^S (open lobe) shows that the wedge loop of SutA slips into the RNAP- β protrusion/lobe gap and wedges RNAP- β lobe open to facilitate promoter loading and unwinding. The structures of SutA-RNAP- σ^S (closed lobe) and RNAP- σ^S holoenzyme suggest that the closed lobe is incompatible with the binding of wedge loop in the RNAP- β protrusion/lobe gap. Based on the structures, we propose that SutA wedge loop opens the main cleft and facilitates the *rrn* promoter

loading and unwinding. The loading of promoter DNA likely displaces SutA-NTD in the main cleft, disengages the wedge loop, and recovers the flexibility of the lobe domain necessary for subsequent promoter unwinding steps (Fig. 7).

- 2. Given the extensive interface observed between σ^S and RNAP, I am surprised that the authors are unable to measure the interactions between σ^S and RNAP using fluorescent polarization experiment. The authors argue that this shows the interactions between σ^S and RNAP is weak. This is in contrast with their cryoEM structure. Further, these negative results raise questions about the sensitivity of the FP experiments and thus the reliability of the FP experiments presented in the manuscript.**

Reply: We are sorry that we didn't state clearly in the manuscript. We measured the binding affinity of the $\sigma^{1.1}$ domain of σ^A or σ^S and RNAP core enzyme, not the binding affinity of σ^A or σ^S and RNAP core enzyme. After weighting the maps of RNAP- σ^S holoenzyme and SutA-RNAP- σ^S complexes, we only observed weak fractionated map signal for $\sigma^{S_{1.1}}$ in the main cleft of RNAP suggesting its conformational heterogeneity and low occupancy (Supplementary Fig. 3). The fluorescence polarization assay failed to detect the weak interaction between $\sigma^{S_{1.1}}$ and RNAP core enzyme, but it showed a strong interaction between $\sigma^{A_{1.1}}$ and RNAP core enzyme (Fig. 2a), consistent with the low occupancy of $\sigma^{S_{1.1}}$ in the cryo-EM structure.

- 3. The authors suggest that the holoenzyme exists in an equilibrium of open and closed β -lobes and SutA binding opens up the β -lobe, widening the downstream DNA binding channel, thus helping with open complex formation. While this is totally plausible, the data here do not support this conclusion as the holoenzyme only shows closed β -lobe conformation (did not show an equilibrium of open and closed β -lobe conformations).**

Reply: Thanks for the comment. Like other bacterial RNAP holoenzyme structures, we only observe one major population of single particles from 3D classification, the reconstructed map from which shows averaged signals for a closed β lobe. Recent molecule dynamics simulation study shows that RNAP- β lobe undergoes rapid oscillations between closed and open conformation¹, likely accounting for observation.

- 4. In the presence of SutA, both the open and closed β -lobe conformations are observed. Can the authors show what differences are there in these two conformations and how SutA binding would promote open β -lobe? Further, the authors show that in the RPo, β -lobe is closed while SutA retains the same interactions with RNAP- σ^S . Again it is unclear how SutA promotes β -lobe opening, thus it is unclear how it promotes open complex formation.**

Reply: In Fig. 3 and Supplementary Fig. 5 of the revised manuscript. We showed that the two conformations differ in the interaction mode of SutA wedge loop, the conformation of β lobe, and the width of main cleft. In the open-lobe conformation, the wedge loop of SutA invades the RNAP- β lobe/protrusion gap, wedges β lobe open, and widens the main cleft. In the closed-lobe conformation, the wedge loop is disordered resulting in a closed lobe and narrow main cleft. Our structures suggest that the conformational differences are apparently attributed to the different interaction modes of SutA wedge loop and that the invasion of SutA wedge loop induces opening of the β lobe domain and widening of main cleft.

- 5. The authors further suggest that the N-terminal acidic regions might compete with $\sigma^{1.1}$ for downstream DNA binding channel, explaining the absence of $\sigma^{1.1}$ density in the SutA bound complex. If this is the case, presumably some complexes will have**

σ 1.1 still in place while others have SutA N-terminal domain occupying this space. Have the authors looked carefully to see if there are two different complexes (classes) in their datasets, one with σ 1.1 in place and the other one with SutA in place?

Reply: We updated the SutA working model in the revised manuscript (Fig. 7). We infer that the major role of N-terminal D/E-rich SutA-NTD is not to displace σ 1.1, but instead to guide the wedge loop into the wedge position. Electrostatic surface presentation of RNAP shows that the main cleft of RNAP is highly positively charged, complementary to the natively charged SutA-NTD (23 D/E residues out of 55 residues in *Pae* SutA). Therefore, SutA-NTD is likely captured in the main cleft and guides the wedge loop across the gap between RNAP- β protrusion and β lobe. Due to intrinsic flexibility of β lobe, the wedge loop slips in the bottom of the protrusion/lobe gap when the β lobe oscillates to its open conformation and locks the open conformation.

Due to lack of secondary structural features and intrinsic flexibility, SutA-NTD likely makes non-specific polar interactions in the main cleft and therefore we do not expect to observe clear map signals of SutA-NTD in the SutA- σ^S -RNAP even it is attracted into the main cleft.

We only observed weak fractionated map signal of $\sigma^S_{1.1}$ in the main cleft of RNAP in the cryo-EM structure of RNAP- σ^S holoenzyme (Fig. 2a and Supplementary Fig. 3a). We propose that $\sigma^S_{1.1}$ exhibits conformational heterogeneity and low occupancy likely due to its low affinity towards RNAP core enzyme. No signal of $\sigma^S_{1.1}$ was observed in both structures of Sut-RNAP- σ^S (Fig. 3a Supplementary Fig. 3a). Both the presence of D/E-rich NTD in the main cleft and the opening of β lobe might interfere with its interaction with RNAP core enzyme.

6. **The authors showed that transcription by RNAP- σ^S and RNAP- σ^A ($\Delta\sigma$ 1.1) could be stimulated by SutA (at very high concentration) while RNAP- σ^A can't. The authors thus suggest that σ 1.1 region is the determination of SutA specificity on RNAP-sigmaS. Exactly how this is achieved is unclear. It would be interesting to elaborate how this is the case based on the structures.**

Reply: Previous reports showed that $\sigma^A_{1.1}$ occupies RNAP main cleft, makes substantial interactions with RNAP- β lobe, and shifts the equilibrium towards the closed conformation^{2,3}. We showed that removal of domain $\sigma^A_{1.1}$ increased its basal transcription activity of $\sigma_{1.1}$, consistent with previous report showing $\sigma^A_{1.1}$ inhibits RPo formation (Fig. 5g)^{4,5}. Moreover, SutA increases the transcription activity of RNAP- σ^A (Δ 1.1) holoenzyme by ~50%, close to the extent by which SutA activates RNAP- σ^S holoenzyme (Fig. 5g). We propose deletion of $\sigma^A_{1.1}$ releases its restrain on RNAP- β lobe and allows SutA wedge loop to slip into the β protrusion/lobe gap and to lock the β lobe in the open conformation.

7. **Further, the authors propose that SutA N terminal domain competes out σ^S , but not σ^A as σ^A σ 1.1 binds tighter to the downstream DNA channel. Given that σ 1.1 has an inhibitory effect on open complex formation, I am surprised that when σ 1.1 is deleted (Figure 5D), in the absence of SutA, the transcription effects are not stimulated. Further, if N-terminal domain of SutA competes out σ^S σ 1.1 in downstream DNA binding channel, SutA should bind to downstream DNA channel more tightly than σ^S , hence would impose higher inhibition for downstream DNA to enter the cleft, thus should be more inhibitory to open complex formation. Taken**

these data together, it is not clear that SutA competing with $\sigma^{1.1}$ in occupying the downstream DNA channel is one of the mechanisms that promotes transcription.

Reply: Thanks for the comments. The data were presented in a misleading way, in which the two experimental serials (wt σ^A and σ^A ($\Delta 1.1$)) were separately normalized with their own controls (SutA=0 nM). In the revised manuscript, all experimental groups were normalized to the same control experimental group (wt σ^A ; SutA=0 nM). The results showed that removal of domain $\sigma^{A_{1.1}}$ increased its basal transcription activity, consistent with previous report showing $\sigma^{A_{1.1}}$ inhibits RPo formation (Fig. 5g)^{4, 5}.

In the revised manuscript, we have updated the working mode of SutA based on the much-improved cryo-EM maps of SutA-RNAP- σ^S structures (Fig. 7). We propose that SutA pinches RNAP- β protrusion with its RNAP-binding domain, and wedges RNAP- β lobe open with its wedge loop and N-terminal D/E-rich region. We propose that the N-terminal D/E-rich loop likely guides the wedge loop into the wedge position but does not directly displace $\sigma^{S_{1.1}}$.

- 8. Overall the data presented here reveal interesting properties of SutA but do not convincingly address the molecular mechanisms of SutA in stimulating σ^S dependent transcription. It would be important for the authors to resolve the functionality of the N-terminus of SutA and to understand how SutA binding induces the conformational changes observed (open β -lobe and dislodging of $\sigma^{1.1}$) in order to reveal the mechanisms of how SutA stimulates transcription. It is possible that with better resolution, the authors might be able to resolve SutA N-terminus in their structures, thus correlating precise SutA binding with the conformation of RNAP- σ^S . What would the structure look like when SutA N-terminal domain is deleted.**

Reply: Thanks for the suggestions. Inspired by the reviewer comments, we have collected a new dataset of SutA-RNAP- σ^S and substantially improved map resolution and model quality of two structures. The improved map allows us to confidently model the RNA-binding domain of SutA, and more importantly, the new map with better resolution allows us to resolve a key loop (wedge loop) of SutA that wedges RNAP- β lobe open in the structure of SutA-RNAP- σ^S (open lobe). The wedge loop is incompatible with the closed lobe and therefore is disordered in the structure of SutA-RNAP- σ^S (closed lobe) (Figs. 3a-d). Based on the new structures, we revised SutA working model. We propose that SutA pinches RNAP- β protrusion with its RNAP-binding domain and wedges open RNAP- β lobe to facilitate promoter unwinding with its wedge loop and N-terminal D/E-rich region.

Reviewer #2 (Remarks to the Author):

He, You et al. have provided structures of the Pseudomonas transcription factor, SutA, with σ S-RNAP. While the observation and conclusions are potentially interesting, there are some red flags on the cryo-EM statistics and analyses. The main new observations are that SutA displaces region 1.1 of σ S more effectively than σ A, explaining its more potent effect on σ S. The second exciting observation is that SutA interacts with the protrusion (previously shown biochemically but shown here structurally), which might lead to a shift in equilibrium to the lobe of RNAP towards the open state. What's unclear and the authors should state so or postulate a mechanism to how that interaction causes the lobe to be more open. Also, it's a paradox that SutA removes region 1.1 of σ S but also then occupies the same channel (seemingly more stably than region 1.1), which would also compete with DNA placement. Can the authors address this?

Reply: We thank the enthusiasm of the reviewer about the unique interaction mode of SutA and RNAP. We also appreciate the comments about the working mechanism of SutA. In the revised manuscript, we have recollected a dataset for SutA-RNAP- σ^S and substantially improved map resolution and model quality of two structures. The resolution of the SutA-RNAP- σ^S (open lobe) has been improved from 3.9 Å to 3.1 Å, and the resolution of the SutA-RNAP- σ^S (closed lobe) has been improved from 4.1 Å to 3.9 Å. The improved map allows us to confidently model the RNA-binding domain of SutA, and more importantly, the new maps with better resolution allow us to resolve a key loop (wedge loop) of SutA that wedges RNAP- β lobe open in the structure of SutA-RNAP- σ^S (open lobe). The wedge loop is incompatible with a closed lobe and therefore is disordered in the structure of SutA-RNAP- σ^S (closed lobe). Based on the new structures, we revised SutA working model (Fig. 7). We propose that SutA pinches RNAP- β protrusion with its RNAP-binding domain and wedges RNAP- β lobe open to facilitate promoter unwinding with its wedge loop and N-terminal D/E-rich region.

We further infer that the N-terminal D/E-rich SutA-NTD guides the wedge loop into the wedge position. Electrostatic surface presentation of RNAP shows that the main cleft of RNAP is highly positively charged (Fig. 3g), complementary to the natively charged SutA-NTD. Therefore, SutA-NTD is likely captured in the main cleft and guides the wedge loop across the gap between RNAP- β protrusion and β lobe. Due to intrinsic flexibility of β lobe, the wedge loop slips in the bottom of the protrusion/lobe gap when β lobe oscillates to its open conformation.

SutA opens the main cleft to facilitate loading and unwinding of *rnn* promoter. After promoter DNA is loaded into the main cleft, it likely displaces the SutA-NTD in the main cleft, disengages the wedge loop, and resumes the flexibility of lobe domain necessary for subsequent promoter unwinding steps.

However, this paper's main detraction is that particle distribution analyses appear to show severe particle orientations in all structures here, with most views unrepresented. Therefore, the authors must present this data using a 3D-FSC analysis. In addition, the authors need meshed density maps and local resolution maps highlighting the features of σ S (region 1) and all of SutA in all structures, so the readers get an idea of the resolution and interpretable densities described in the manuscript. Supplementary Fig 5D is very strange- the local resolution goes from 7 Å to 4 Å without anything in between? Are the authors sure they did this analysis correctly?

Reply: Regarding the maps. We included CHAPSO during cryo-EM sample preparation and therefore avoided particle orientation bias⁶. We have calculated the 3D-FSC curves for all maps as suggested and angular distribution plots for all data (Supplementary Fig. 2, 4, and 6). The analysis showed no obvious particle orientation bias for all datasets. The sphericity values for RNAP- σ^S holoenzyme, SutA-RNAP- σ^S (open lobe), SutA-RNAP- σ^S (closed lobe), and SutA- σ^S -RPO are 0.754, 0.981, 0.921, and 0.941, respectively.

We have prepared the new Supplementary Fig. 3 to show side-by-side comparison of the weighted density map for $\sigma_{1.1}^S$ and SutA as suggested.

The resolution grid setting in Relion was inappropriate to calculate the local resolution map of SutA-RPO in the previous manuscript. The new figure (Supplementary Fig. 6d) shows a reasonable local resolution distribution of SutA-RPO map.

Minor comments are listed below:

- 1. Pg3 ln 71: this is incorrect, according to the cited paper: '(SutA) activates transcription from the rrn promoter by both the housekeeping sigma factor holoenzyme (E σ 70) and the stress sigma factor holoenzyme (E σ S) in vitro, but has a greater impact on E σ S.'**

Reply: Thanks for the correction. We have reworded the sentence in the revised manuscript.

- 2. Pg4 ln 106: Specificity for σ^S is questionable; also binds σ^A (resulting in rrn upregulation)**

Reply: Thanks for the correction. The sentence has been reworded as below

“SutA is unique in its large portion of disordered regions, highly structural flexibility, and RNAP- σ^S holoenzyme preference.”

- 3. C-term truncation of SutA appears to incompletely suppress the impact of SutA on transcription in Fig1C assay; this should be mentioned.**

Reply: Thanks for the comment. We have added the following sentence in the revised manuscript.

“In contrast, the short C-terminal tail of SutA contributes less to the transcription activation activity (Fig. 1c, left)”

- 4. Raw fluorescence values or images of the mango riboswitch in operation would help to illustrate the effect in Fig 1C**

Reply: We have prepared Supplementary Fig. 1e to illustrate the experiment setup.

- 5. Claiming that the visual absence of sigma 1.1 in SutA+Holo structure indicates 'incompatibility between Sigma 1.1 and SutA' is not necessarily supported. Sigma 1.1 might just be freer to move around in the SutA+Holo structure due to overall clamp-lobe orientation- should be stated as an alternative hypothesis. Also, one would need to have the two maps with the same levels and contour and a close-up view. Pymol can weigh the maps similarly. Also, if SutA can outcompete $\sigma_{1.1}$, why don't we see it in the map (we see $\sigma_{1.1}$). The authors should address that.**

Reply: Thanks for the comment and useful suggestion for weighting maps. We have recollected a dataset of SutA-RNAP- σ^S and the improved map resolution allows us to resolve the wedge loop that is jammed in the RNAP- β protrusion/lobe gap to open the β lobe for loading of promoter DNA (Fig. 3). We agree that the wedge loop and D/E-rich

NTD don't necessarily displace $\sigma_{1.1}^S$, but likely only increase the flexibility of $\sigma_{1.1}^S$ that makes it easier to be displaced during promoter loading and unwinding.

Due to lack of secondary structural features and intrinsic flexibility, SutA-NTD likely makes non-specific polar interactions in the main cleft and therefore we do not expect to observe clear map signals of SutA-NTD in the two structures of SutA- σ^S -RNAP.

We have weighted all map in Pymol as suggested and prepared a figure (Supplementary Fig. 3) to show side-by-side comparison of the weighted density map for $\sigma_{1.1}^S$ and SutA.

6. The fluorescence anisotropy assay might require some troubleshooting to see σ S 1.1 binding, as the cryo-EM maps suggest it must bind RNAP to some detectable extent.

Reply: After we weighted cryo-EM maps for all structures, we showed that the map signals for $\sigma_{1.1}^S$ is very weak and fractionated (Supplementary Fig. 3a), suggesting its conformational heterogeneity and low occupancy. The structure is consistent with the fluorescence polarization results.

7. 4096 nM is a lot of SutA; I'd be cautious in using this to determine biologically relevant binding tendencies of SutA to holo.

Reply: Thanks for the comment. We have removed the SutA competition curves in the revised manuscript. The new Fig. 2C only includes the binding affinity curves of $\sigma_{1.1}^S$ and $\sigma_{1.1}^A$.

8. 'the structural and biochemical evidence suggests that SutA-RBD anchors to RNAP- β protrusion and SutA-NTD invades the main cleft to open RNAP- β lobe and release σ S1.1 from the main cleft.' I would caveat this claim; SutA-NTD is nowhere to be seen in the maps.

Reply: Thanks for the comment. We have removed the description in the revised manuscript.

9. The idea that SutA-NTD invades the cleft, widens the clamp-lobe spacing, and displaces σ A 1.1 to thereby facilitate DNA loading is problematic. It is claimed that σ S 1.1 binds so weakly that fluorescence polarization assays can't detect the binding, and yet SutA-NTD, which successfully outcompetes and displaces the far more challenging σ A 1.1, is supposed to be an easier obstacle for DNA loading to overcome than σ S 1.1? This is inconsistent with the proposed model.

Reply: We updated the SutA working model in the revised manuscript (Fig. 7). We infer that the major role of N-terminal D/E-rich SutA-NTD is not to displace $\sigma_{1.1}$, but instead to guide the wedge loop into the wedge position. SutA-NTD is likely captured in the main cleft and guides the wedge loop across the gap between RNAP- β protrusion and β lobe. Due to intrinsic flexibility of β lobe, the wedge loop slips in the bottom of the protrusion/lobe gap when the β lobe oscillates to its open conformation and locks the open conformation. SutA wedge loop opens the main cleft and facilitates loading and unwinding of the *rrn* promoter. After promoter DNA has been loaded into the main cleft, it likely displaces SutA-NTD from the main cleft, disengages the wedge loop, and resumes the flexibility of β lobe necessary for subsequent promoter unwinding steps.

10. Description of the stopped-flow assay and how it illustrates RPO formation wasn't very clear.

Reply: We have added the highlighted words in the revised manuscript as below,

“we measured the kinetics of RPo formation by a stopped-flow fluorescence assay, in which the Cy3 fluorophore (attached to the +2 position of the non-template strand of promoter DNA) serves as a probe sensing local environment **change and increases its fluorescence upon promoter unwinding**”.

11. **Fig 5D, are these the same assays as 1C- if so, please state in legend? Also, it would be good to illustrate this effect with σ S 1.1 as well (full length and 1.1 truncations). Conclusions from this assay are inconsistent with the paper's earlier remarks (line 71). Loss of this effect for σ S and σ A 1.1 regions where truncated SutA (missing NTD) is used would help to corroborate the paper's model**

Reply: We have added the highlighted words in the figure legend of revised manuscript as below to illustrate the stopped-flow fluorescence assay.

“**The results of fluorescence-based in vitro transcription assay show that SutA increases *rrn* promoter transcription by RNAP- σ^A or RNAP- σ^A (Δ 1.1) holoenzymes**”

Because we have revised working model of SutA and the new model shows that the major role of N-terminal D/E-rich SutA-NTD is not to displace σ 1.1, but instead to guide the wedge loop into the wedge position. Although σ 1.1 itself is not the focus of the revised manuscript, we still keep the experimental results of RNAP- σ^A (Δ 1.1), which support the new model and show that deletion of σ^A 1.1 releases its restraint on RNAP- β lobe and the increased conformational flexibility of β lobe allows SutA wedge loop to slip into the β protrusion/lobe gap, lock the β lobe in an open conformation, and facilitate promoter unwinding.

12. **Line 263, the word 'observation' is inappropriate here as there is no SutA-NTD visible in these maps (as far as this figure shows).**

Reply: We have removed the sentence in the revised manuscript.

References

1. Unarta IC, *et al.* Role of bacterial RNA polymerase gate opening dynamics in DNA loading and antibiotics inhibition elucidated by quasi-Markov State Model. *Proc Natl Acad Sci U S A* **118**, (2021).
2. Basu RS, *et al.* Structural basis of transcription initiation by bacterial RNA polymerase holoenzyme. *J Biol Chem* **289**, 24549-24559 (2014).
3. Zuo Y, Wang Y, Steitz TA. The mechanism of E. coli RNA polymerase regulation by ppGpp is suggested by the structure of their complex. *Molecular cell* **50**, 430-436 (2013).
4. Shin Y, Qayyum MZ, Pupov D, Esyunina D, Kulbachinskiy A, Murakami KS. Structural basis of ribosomal RNA transcription regulation. *Nat Commun* **12**, 528 (2021).
5. Chen J, *et al.* E. coli TraR allosterically regulates transcription initiation by altering RNA polymerase conformation. *Elife* **8**, (2019).
6. Chen J, Noble AJ, Kang JY, Darst SA. Eliminating effects of particle adsorption to the air/water interface in single-particle cryo-electron microscopy: Bacterial RNA polymerase and CHAPSO. *J Struct Biol X* **1**, (2019).

Reviewers' Comments:

Reviewer #1:

Remarks to the Author:

The results and manuscript have much improved due to the improved resolution of the cryoEM reconstructions which enabled the visualisation of SutA in the b'-lobe region. Specifically, they observe density that is inserted in-between the b'lobe and b'protrusion, thus causing the opening of the cleft. Although the density is of insufficient resolution to reveal molecular details, the data are clear that a closed cleft conformation is incompatible with the insertion of SutA peptide/loop in this region. The revised model now is consistent with previous data and those presented here. The revised manuscript has thus addressed almost all my concerns and criticisms and can be accepted for publication after minor revisions that will clarify the conclusions:

1) Figure 3 clearly shows there is density for the SutA wedge loop that is inserted between b-lobe and b protrusion. Although the resolution is probably insufficient to reveal molecular details, it would be informative to show the density in greater details and show a polyaniline model in that region. Figure 3C insert could be better utilised to illustrate this point. It will show greater details about the interactions between SutA and b-subunit and helps explaining how SutA helps/promotes b-lobe opening. It would be useful to show how the closed-cleft conformation is incompatible with SutA wedge loop insertion in Figure 3d.

2) It would be helpful in the final figure to have a direct comparison of the RNAP-sigmaS, RNAP-sigmaS-SutA open/closed conformation and SutA-RPo structures, with similar views as in Figure 3 where it clearly displays the SutA density and the cleft opening.

3) the model in Figure 7 should use similar views and depictions as in Figure 3 with the b-strand and a-helix of SutA in similar positions, but the wedge loop insertion is correlated with cleft opening. Presumably once DNA is loaded, cleft will close to stabilise the open complex. The cleft closure would then force the wedge loop of SutA to be relocated.

Reviewer #2:

Remarks to the Author:

He, You et al. have provided a much-improved manuscript describing the structural and biochemical basis of activation by the Pseudomonas transcription factor, SutA, on σ S-RNAP. The resubmitted manuscript contains improved structural data that led the authors to amend their model. The new observations are that SutA binds the protrusion domain of RNAP (previously shown biochemically but shown here structurally and genetically), inserts a wedge loop between the lobe and protrusion, which leads to widening the main cleft. The authors propose that this stabilization of the open cleft facilitates open complex formation by making room for the DNA to load and unwind. They provide biochemical data showing that SutA increases open complex formation, likely due to their proposed mechanism. The manuscript is more rigorous and scholarly, and the authors present sufficient structural and biochemical validation of the proposed mechanism of this new and unusual transcription factor. It's becoming apparent that new regulatory mechanisms and insight into fundamental processes are emerging as new factors or RNAPs are being included in biophysical investigations. The revised manuscript can be accepted with only one suggestion being incorporated. They should note that an earlier paper that described the structure of Mtb RNAP with myxopyronin also suggested that the lobe must move for the DNA to load and unwind as myxopyronin freezes the clamp (the other pincer) and yet, the DNA loads most of the way (reference 12). It is important to state the rationale behind the original proposal because, as the authors note, the view in the field was that the clamp was the only mobile element required to open for promoter melting, and ref 12 provided the first evidence that that might not be true. What's exciting is that THIS manuscript now presents data describing that the unusual transcription factor, SutA, stabilizes that necessary motion.

Reply to reviewers' comments:

Reviewer #1 (Remarks to the Author):

The results and manuscript have much improved due to the improved resolution of the cryoEM reconstructions which enabled the visualisation of SutA in the β' -lobe region. Specifically, they observe density that is inserted in-between the β' lobe and β' protrusion, thus causing the opening of the cleft. Although the density is of insufficient resolution to reveal molecular details, the data are clear that a closed cleft conformation is incompatible with the insertion of SutA peptide/loop in this region. The revised model now is consistent with previous data and those presented here. The revised manuscript has thus addressed almost all my concerns and criticisms and can be accepted for publication after minor revisions that will clarify the conclusions:

1. **Figure 3 clearly shows there is density for the SutA wedge loop that is inserted between β -lobe and β protrusion. Although the resolution is probably insufficient to reveal molecular details, it would be informative to show the density in greater details and show a polyalanine model in that region. Figure 3C insert could be better utilized to illustrate this point. It will show greater details about the interactions between SutA and b-subunit and helps explaining how SutA helps/promotes b-lobe opening. It would be useful to show how the closed-cleft conformation is incompatible with SutA wedge loop insertion in Figure 3d.**

Reply: Thanks for the suggestion. We have revised the inserts of Fig. 3a and 3c. The new inserts show map details of the wedge loop and the nearby structure regions of β lobe and β protrusion. We have also revised Fig. 5d to include the potential residues that are likely involved in the interaction between the wedge loop of SutA and RNAP- β subunit. As Fig. 5e has already shows the steric clash of the wedge loop of SutA and RNAP- β subunit, we did not make additional changes to Fig. 3c.

2. **It would be helpful in the final figure to have a direct comparison of the RNAP-sigmaS, RNAP-sigmaS-SutA open/closed conformation and SutA-RPo structures, with similar views as in Figure 3 where it clearly displays the SutA density and the cleft opening.**

Reply: Thanks for the suggestion. We prepared a Supplementary Fig. 7 to directly compare the cryo-EM maps of *Pae* RNAP- σ^S , SutA-RNAP- σ^S (open lobe), SutA-RNAP- σ^S (closed lobe), and SutA- σ^S -RPo structures at the same view orientation.

3. **Th model in Figure 7 should use similar views and depictions as in Figure 3 with the b-strand and a-helix of SutA in similar positions, but the wedge loop insertion is correlated with cleft opening. Presumably once DNA is loaded, cleft will close to stabilise the open complex. The cleft closure would then force the wedge loop of SutA to be relocated.**

Reply: Thanks for the suggestion. We have revised the models in Fig. 7 to better illustrate the conformational change of SutA wedge loop and β lobe in the four structures.

Reviewer #2 (Remarks to the Author):

He, You et al. have provided a much-improved manuscript describing the structural and biochemical basis of activation by the *Pseudomonas* transcription factor, SutA, on σ^S -RNAP. The resubmitted manuscript contains improved structural data that led the

authors to amend their model.

The new observations are that Suta binds the protrusion domain of RNAP (previously shown biochemically but shown here structurally and genetically), inserts a wedge loop between the lobe and protrusion, which leads to widening the main cleft. The authors propose that this stabilization of the open cleft facilitates open complex formation by making room for the DNA to load and unwind. They provide biochemical data showing that Suta increases open complex formation, likely due to their proposed mechanism. The manuscript is more rigorous and scholarly, and the authors present sufficient structural and biochemical validation of the proposed mechanism of this new and unusual transcription factor. It's becoming apparent that new regulatory mechanisms and insight into fundamental processes are emerging as new factors or RNAPs are being included in biophysical investigations. The revised manuscript can be accepted with only one suggestion being incorporated. They should note that an earlier paper that described the structure of Mtb RNAP with myxopyronin also suggested that the lobe must move for the DNA to load and unwind as myxopyronin freezes the clamp (the other pincer) and yet, the DNA loads most of the way (reference 12). It is important to state the rationale behind the original proposal because, as the authors note, the view in the field was that the clamp was the only mobile element required to open for promoter melting, and ref 12 provided the first evidence that that might not be true. What's exciting is that THIS manuscript now presents data describing that the unusual transcription factor, Suta, stabilizes that necessary motion.

Reply: Thanks for pointing out the earlier report about the lobe movement. We have included the following words in the discussion section.

“A previously reported cryo-EM structure of coralloyronin A-bound Mycobacterium tuberculosis transcription initiation complex showed that the promoter DNA passed the β gate loop- $\sigma_{1,2}$ obstacle and occupied the main cleft in a partially melted form ¹². It is likely that the rotation of RNAP- β lobe accounts for opening of the main cleft to allow entry of the promoter DNA when the mobility of RNAP- β ' clamp is inhibited by coralloyronin A.”.